# ABC: Achieving Better Control of Visual Embeddings using VLLMs

**Benjamin Schneider,** *University of Waterloo*      *Benjamin.Schneider@uwaterloo.ca*

**Florian Kerschbaum,** *University of Waterloo*      *florian.kerschbaum@uwaterloo.ca*

**Wenhu Chen,** *University of Waterloo*      *wenhuchen@uwaterloo.ca*

**Reviewed on OpenReview:** *https://openreview.net/forum?id=RezANmBpxW*

## Abstract

Visual embedding models excel at zero-shot tasks like visual retrieval and classification. However, these models cannot be used for tasks that contain ambiguity or require user instruction. These tasks necessitate an embedding model which outputs can use a natural language instruction to control the representation of a visual embedding. Existing CLIP-based approaches embed images and text independently, and fuse the result. We find that this results in weak interactions between modalities, and poor user control over the representation. We introduce **ABC**, an open-source multimodal embedding model that uses a vision-language model backbone to deeply integrate image features with natural language instructions. **ABC** achieves best-for-size performance on MSCOCO image-to-text retrieval and is the top performing model on classification and VQA tasks in the Massive Multimodal Embedding Benchmark. With a strongly unified vision-language representation, **ABC** can use natural language to solve subtle and potentially ambiguous visual retrieval problems. To evaluate this capability, we design `CtrlBench`, a benchmark that requires interleaving textual instructions with image content for correct retrieval. **ABC** advances the state of visual embeddings, outputting high-quality visual representations with natural language control. Our model and datasets are available at our project page:

https://tiger-ai-lab.github.io/ABC/

## 1 Introduction

Visual embeddings have become a foundational representation in computer vision. Image embedding models have become the state of the art for many zero-shot tasks, including visual retrieval (Chen et al., 2024) and image classification (Yu et al., 2022). Since the release of CLIP (Radford et al., 2021), its dual encoder architecture has remained the state of the art for producing high-quality visual embeddings (Yu et al., 2022; Sun et al., 2023; Chen et al., 2024). However, CLIP only supports *separately* embedding images or text (Radford et al., 2021). Therefore, complex visual embedding tasks which require additional specification are impossible. For example, a CLIP model cannot distinguish which is the best answer in Figure 1. All candidates are plausible unless the user provides additional instruction. However, additional conditioning on a natural language question *disambiguates* the query, making it apparent that the bottom candidate is the best answer (Stelmakh et al., 2022).

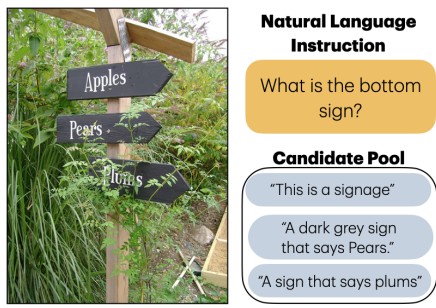

Figure 1: A task from `CtrlBench`. Conditioning on the natural language instruction is required to unambiguously retrieve a candidate.

Conditioning on natural language gives the user greater control over their visual query. This is especially critical for retrieval augmented generation systems, which must unambiguously retrieve a small number of highly relevant facts from large knowledge bases, to support accurate text generation (Amiraz et al., 2025).

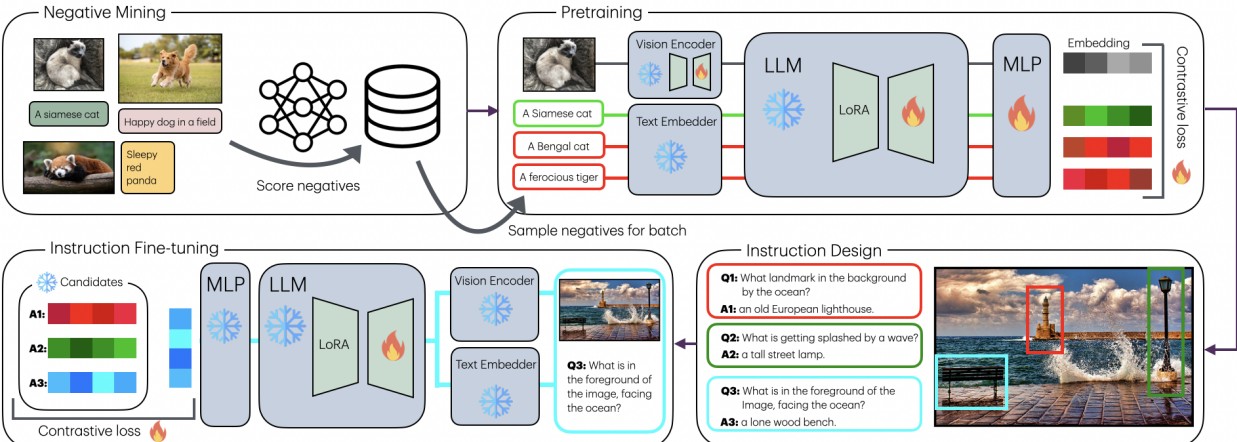

Figure 2: An overview of **ABC**'s training regime. First, we mine negatives to augment each batch in our pretraining dataset with *almost plausible* text candidates. Then we pretrain our model using contrastive loss on independently embedded text-image pairs. Once our model has learned to output high-quality uncontrolled visual embeddings, we introduce instructions. We create multiple instructions and candidate captions for a single image, with each one focusing on different aspects of the image. Lastly, we fine-tune using multiple captions for same image, the positive caption for each query corresponds to the natural language instruction. Therefore, **ABC** uses the instruction to distinguish the correct candidate for the query.

We find that existing approaches suffer from two problems: **(1)** Reliance on generic or human-annotated instructions (Jiang et al., 2024; Zhang et al., 2025). Unlike the abundant noisy image-text data used to train CLIP models Schuhmann et al. (2022); Changpinyo et al. (2021); Sharma et al. (2018), datasets containing multiple modalities and diverse instruction phrases are significantly more scarce and challenging to curate at scale. This is problematic for embedding models, which require massive datasets for the best representations (Cherti et al., 2023a). Often vague instructions are used repetitively during training, resulting in overfitting of the instruction (Wei et al., 2023; Gudibande et al., 2023). **(2)** Weak interaction between modalities. Previous works fuse embeddings outputted by CLIP models Zhang et al. (2024); Wei et al. (2023). This approach prevents deeper interaction between modalities, resulting in superficial use of the instructions Jiang et al. (2024).

To this end, we introduce **ABC**, a model that uses natural language instructions to control visual embeddings. **ABC**'s Vision Large language Model (VLLM) backbone allows it to integrate natural language instructions when crafting visual embeddings. We find that training our model has two fundamental challenges: **(1)** Extracting useful contrastive embeddings from a pretrained generative VLLM. **(2)** Designing an instruction fine-tuning method that lets users modify multimodal embeddings using natural language instructions. To train **ABC**, we adopt a multi-stage training process. In the initial pretraining stage, we use contrastive training with carefully selected negatives to develop a model that generates embeddings, similar to CLIP. In the instruction fine-tuning stage, we train a lightweight adapter to project queries consisting of images and natural language instructions into the embedding space of the pretrained model. Our synthetic instructions to correspond to different aspects *of the same image*, therefore, **ABC** learns to use the instruction for fine-grained control. Our training results in a model that produces powerful and controllable visual embeddings.

**ABC** achieves impressive zero-shot performance in retrieval, classification, and visual question answering (VQA) tasks. In MSCOCO (Lin et al., 2015) image-to-text retrieval, our model outperforms all CLIP models containing at most 8 billion parameters. Furthermore, our model outperforms all other models on the zero-shot classification and VQA splits of MMEB (Jiang et al., 2024), a multimodal embedding benchmark spanning 19 tasks. Lastly, we design `CtrlBench` to measure our model's ability to use natural language instructions to control retrieval. Using `CtrlBench`, we show that **ABC** can accomplish visual retrieval tasks that are fundamentally ambiguous without utilizing natural language instructions.

Our contribution is threefold. **(1) ABC**: an open-source multimodal embedding model that uses natural language instructions to control visual embeddings. We demonstrate that **ABC** produces powerful embeddings

by benchmarking on zero-shot tasks *and* robustly utilizes natural language instructions to control embeddings. We study how key design choices—including vision encoder resolution, contrastive loss temperature, and VLLM backbone—impact embedding quality. **(2)** We provide a *decoupled* methodology for adapting VLLMs into dense embedding models. Previous work integrates instructions during the large-scale contrastive training run. We show that these stages can be decoupled, resulting in a lightweight and adaptable fine-tuning stage that requires only 100 training steps using synthetic instructions. **(3)** Lastly, we introduce `CtrlBench`, a novel benchmark for measuring instruction-controlled retrieval. `CtrlBench` *requires* the model to interleave modalities to retrieve the correct response.

## 2 Background

**Visual embeddings.** Radford et al. (2021) proposed CLIP, which projects images and text to a shared dense embedding space. Due to its contrastive training, the similarity of vectors is computed as simple dot product (Hadsell et al., 2006). CLIP embeddings are used for tasks such as zero-shot image classification and image-to-text retrieval, by computing the similarity of an image query $q$ with a set of candidates $c \in \mathcal{C}$. An embedding model $f$ maps queries and candidates to vectors in $\mathbb{R}^n$. Given a query $q$ and a set of candidates $\mathcal{C} \subseteq \mathbb{R}^n$ containing the correct answer $c^+$, the retrieved candidate is:

$$c^* = \arg\max_{c \in \mathcal{C}} \ f(q) \cdot f(c)$$

The model is succeeds if $c^* = c^+$. Alternatively, when measuring metrics such as recall@K, K of the top scoring candidates are returned. Subsequent works have identified several key factors for training training high-quality CLIP models: DataComp (Gadre et al., 2023) demonstrated the importance of data choice during to create high-quality embeddings. (Cherti et al., 2023b) showed that large-scale pretraining and batch size are crucial for producing the best performing models. Lastly, Align (Jia et al., 2021) found that optimizing the setting of the temperature $\tau$, used to scale the contrastive loss function, was crucial.

**Multimodal embeddings for Visual Question Answering.** Visual embeddings, especially those of diverse scenes containing many objects, offer limited control over the embedding (Wei et al., 2023). However, a multimodal embedding $q = (q_i, q_t)$, jointly conditioned on image $q_i$ and text instruction $q_t$ allows for flexible user natural language control over embeddings. This allows the user to create embeddings corresponding to specific objects or concepts in an image (Jiang et al., 2024). For a VQA task, the model embeds $(q_i, q_t)$ jointly into a dense vector. The model uses cosine similarity to retrieve the maximally similar text candidate $c^* \in \mathcal{C}$. As in the unconditioned setting, the model is successful if $c^*$ the correct answer. We use the VQA split of MMEB (Jiang et al., 2024), which contains 10 tasks that span text-conditioned classification and retrieval. We also introduce `CtrlBench`, a benchmark designed to measure text-conditioned visual retrieval when image-only retrieval is *ambiguous*.

**Ambiguity in visual retrieval.** Ambiguity is a well studied problem in information retrieval and natural language processing (Keyvan & Huang, 2022). Often, users provide queries that have multiple distinct, nevertheless plausible answers. Several rectifying approaches have been proposed, including using follow-up questions to disambiguate queries (Kim et al., 2023; Aliannejadi et al., 2019) and query refinement (Stelmakh et al., 2022). Following Min et al. (2020), we define an image query $q_i$ as ambiguous if its corresponding candidate pool $\mathcal{C}$ contains multiple semantically distinct plausible answers. Additional conditioning, such as on a natural language instruction $q_t$, disambiguates $q_i$ if the joint query $q = (q_i, q_t)$ has exactly one plausible answer in $\mathcal{C}$ (Stelmakh et al., 2022).

**Extracting Embeddings from LLMs.** Generative large language models are a source of pretrained bases training dense embeddings models. NV-Embed (Lee et al., 2024) used contrastive training on an LLM backbone to achieve the best performance on the MMEB text embedding benchmark (Muennighoff et al., 2023). Several architectural modifications have been proposed for adapting LLMs to produce dense embeddings Wang et al. (2024a); BehnamGhader et al. (2024); Lee et al. (2024). LLM2VEC BehnamGhader et al. (2024) changed the attention mask to be bidirectional to allow information flow between all tokens. Furthermore, several methods for extracting dense embedding from LLMs hiddens have been proposed. These include picking the last token (Jiang et al., 2024), mean token pooling (BehnamGhader et al., 2024) or an additional attention layer (Lee et al., 2024).

## 3 ABC

Figure 2 is an overview of our training regime and model architecture. Our training regime consists of 2 distinct stages; pretraining and instruction fine-tuning. The pretraining stage trains on image-caption pairs to adapt features used for generative modeling into features for dense embeddings. As our pretraining does not require instructions, it can be easily scaled using any large image-captioning data source (Changpinyo et al., 2021; Schuhmann et al., 2022). In the second stage, we train an adapter to allow **ABC** to embed images conditioned on a text instruction, enabling natural language control over visual representations. Using contrastive learning with synthetic instructions, the **ABC** learns to represent different visual aspects of the same image based solely on textual commands.

### 3.1 Model Design

**Pretraining Architecture**. During pretraining images and text are embedded independently, and aligned using contrastive loss, much like CLIP (Radford et al., 2021). Text and images are converted to hiddens $\boldsymbol{h}^{(0)} \in \mathbb{R}^{s \times d}$ using the embedding layer and vision encoder, respectively. The LLM module is shared between both modalities. To adapt the LLM to output dense embeddings, we make several architectural changes. Following BehnamGhader et al. (2024), we enable bidirectional attention, allowing all tokens to attend to all other tokens. To create our dense embeddings, we average over the sequence dimension in the last hidden layer $\boldsymbol{h}^{(l)}$ (equation 1), and project the result $\boldsymbol{x} \in \mathbb{R}^d$ using a residually connected $MLP$ layer (equation 2).

$$\boldsymbol{x} = \frac{1}{s} \sum_i^s \boldsymbol{h}_i^{(l)} \tag{1}$$

$$MLP(\boldsymbol{x}) = \boldsymbol{x} + \boldsymbol{A}g(\boldsymbol{B}\boldsymbol{x}) \tag{2}$$

Where $\boldsymbol{A}$ and $\boldsymbol{B}$ are parameter matrices and $g$ is the element-wise SELU function (Klambauer et al., 2017). To train our backbone, we apply LoRA (Hu et al., 2021) adapters on both the vision encoder and LLM modules. We optimize the contrastive loss temperature hyperparameter $\tau$ during pretraining.

**Instruction Fine-tuning Architecture**. After pretraining, **ABC** can project queries consisting of images *or* text into a shared embedding space. However, to control the representation of visual embeddings, we must project queries consisting of images *and* text instructions into the embedding space. To accomplish this, we train a low-rank LoRA adapter on the LLM component. The LoRA hooks of the instruction adaptor are attached to the MLP and attention layers in each decoder block of the LLM backbone, we freeze all other weights during instruction fine-tuning. The adapter contrastively aligns image and instruction queries with text candidates embedded using the pretrained model. For queries containing both an image and text instruction, we adopt the instruction template from our VLLM backbone (Wang et al., 2024b):

$$\texttt{<vision\_start> <image\_tokens> <vision\_end> <im\_start> Instruction: \{text\} <im\_end>} \tag{3}$$

We replace `<image_tokens>` with tokens output by vision encoder. `<vision_start>` and `<vision_end>` are special vocabulary tokens used to denote the start and end of the sequence of image tokens. Similarly, `<im_start>` and `<im_end>` are used to denote the start and end of text tokens. If no natural language instruction is used, the last token is `<vision_end>`. We do not back-propagate through text candidate embeddings, only the queries containing an image and instruction. This ensures that our image-text queries share the same features as text and images embedded using the pretrained model. We can easily alternate between embedding with or without instructions by disabling the instruction fine-tuned LoRA adapter. We freeze the temperature $\tau$ during fine-tuning.

### 3.2 Pretraining Data

To create our pretraining dataset we employ negative mining on Conceptual Captions (Sharma et al., 2018). We derive the mined negatives for our dataset as follows: **(1)** We do a small pretraining run using only in-batch negatives. **(2)** We use the resulting model to calculate similarity scores between all images and captions in our pretraining dataset. This approach avoids a circular dependence on a third-party embedding model to train our embedding model. Therefore, it is easily extensible to modalities where an existing embedding model is not publicly available. **(3)** To prevent our negatives from being too similar to our positive samples, we set a similarity threshold $\epsilon \in [0, 1]$. We only sample negatives that have a similarity score of at most $\epsilon$ times the similarity score of the correct candidate. We randomly choose our mined hard negatives from the 100 candidate captions below the threshold. This results in the text candidates shown in Figure 3. The mined text negatives are clearly relevant, but the correct caption is still the best answer.

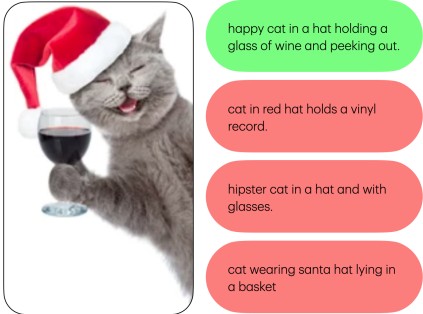

Figure 3: The positive caption (green) is the best caption for the image. The mined negatives (red) are relevant but not the best choice.

In our pretraining run, each batch consists of $N$ image queries (without instructions) and $M$ text candidates. We include $M - N$ mined text negatives in each batch. Therefore, each image query has $\frac{M}{N} - 1$ corresponding mined negatives. Our pretraining loss function is given by Equation (4).

$$\mathcal{L}_{pretrain} = -\sum_{i}^{N} \log \frac{\exp(f(\boldsymbol{q}_i) \cdot f(\boldsymbol{c}_i^+)/\tau)}{\sum_{j=1}^{N}(\exp(\frac{f(\boldsymbol{q}_i) \cdot f(\boldsymbol{c}_j^+)}{\tau}) + \sum_{k=1}^{\frac{M}{N}-1} \exp(\frac{f(\boldsymbol{q}_i) \cdot f(\boldsymbol{n}_j^k)}{\tau}))} \tag{4}$$

For a given query $\boldsymbol{q}_i$, $\boldsymbol{n}_i^k$ is the $k_{th}$ mined hard negative candidate and $\boldsymbol{c}_i^+$ is its positive candidate. $\tau$ is the temperature hyperparameter used to scale the loss. $\boldsymbol{n}_j^k$ is a mined negative for $\boldsymbol{x}_i$ when $i = j$, otherwise it acts as a regular in-batch negative. The embeddings are unit normalized before loss is computed. We scale the loss by $\frac{1}{N}$, the number of image queries in the batch.

### 3.3 Instruction Fine-tuning Data

To create our instruction fine-tuning dataset we use Visual Genome (Krishna et al., 2016), a dataset comprised of images with captioned bounding boxes. When choosing which bounding boxes to use for each image, we filter by the size of the bounding box (width × height). We exclude the 5 largest bounding boxes, as we find they often do not represent specific objects or aspects of the scene. For each image, we randomly choose 4 bounding boxes and their respective captions. We then prompt GPT-4o (OpenAI, 2024) to create instructions corresponding to each bounding box caption. We sample multiple captions from each image so that they can be used as negatives for each other during instruction fine-tuning. This ensures that each image query has multiple realistic captions in its batch. Therefore, *utilizing the instruction is required to choose the correct text candidate unambiguously.* We provide the prompt and generation settings used to create our instruction fine-tuning dataset in Appendix A. We instruction fine-tune using exclusively in-batch negatives. However, we group all queries that contain the same image into the same batch. This ensures that queries always have multiple relevant text candidates.

## 4 Experiments

We evaluate two aspects of the model. As controlling a poor-quality embedding is useless, we first assess the quality of embeddings output by **ABC**. We evaluate our model on a variety of zero-shot retrieval and classification tasks (Section 4.3). We study how variable resolution capabilities (Section 4.4), the settings for $\tau$ when contrastively training a VLLM (Section 4.5) and VLLM backbone choice (Section 4.6) effect embedding quality. Secondly, we measure **ABC**'s ability to use natural language to control visual embeddings on several visual question answering tasks (Section 4.7). Motivated by shortcomings in VQA benchmarks, we construct

`CtrlBench`. A benchmark designed to require the model to jointly use the image and instruction to retrieve the most appropriate caption.

## 4.1 Training Settings

We initialize **ABC** using Qwen2-VL-7B (Wang et al., 2024b). We train our negative mining model using only in-batch negatives with a batch size of 256 for 1000 steps. We set $\epsilon = 0.95$ and sample 7 mined negatives for each image. Mined negatives are sampled across the pretraining dataset. We pretrain using batches of 512 image queries and 4096 text candidates sharded across 8 NVIDIA A100-SXM4-80GB GPUs (Qu et al., 2021) for 4000 steps. We use a LoRA adapter with a rank of 64 and a fixed alpha of 128. We limit the number of tokens output by the vision encoder to 512 during training. For our optimizer, we use AdamW (Loshchilov & Hutter, 2019) with a learning rate of $4 \times 10^{-5}$, betas of 0.9 and 0.999 and a weight decay of $10^{-3}$. We warm-up for 3% of training steps and initialize the temperature $\tau$ as $7 \times 10^{-2}$. In our instruction fine-tuning stage, we use a lower rank LoRA adapter. We set the rank and alpha to 16 and 32, respectively. Our instruction fine-tuning stage can be short, as our VLLM backbone is already instruction fine-tuned. Therefore, we only instruction fine-tune for 100 steps. Each batch contains 128 unique images, with each image appearing four times, paired with a different instruction and a corresponding positive text candidate.

**Modifications to save VRAM.** Due to our use of LoRA Hu et al. (2021), the VRAM requirements for gradients and optimizer state is relatively low. Consequentially, the model activations used by autograd for the backward pass account for most of VRAM used during training. To address this, we make aggressive use of activation checkpointing, recomputing the activations of each decoder block during the backward pass. Furthermore, we modify the VLLM backbone to skip the logits and cross-entropy loss computation. With the large vocabulary size of modern LLMs, the output layer has become the most memory-intensive layer (Wijmans et al., 2024). As we only use the hidden state for calculating embeddings, the logits tensor is unnecessary. We find that this simple change saves up to 11 GB of memory per device, allowing us to use to a significantly larger batch size.

## 4.2 Inference Settings

When evaluating ABC we adjust the vision encoder to output a number of tokens corresponding to the resolution of the image. We rescale images to the nearest resolution that fits our vision encoder, which has a $14 \times 14$ pixel patch size (Wang et al., 2024b). For very high resolution images, we cap the number of visual tokens used to encode an image to 1024, down-scaling the image accordingly. When evaluating models that supports variable resolution, we vary their token budgets similarly (Jiang et al., 2024; Zhang et al., 2025). However, when we attempt to increase token budgets in CLIP models using the same method, we find that it degrades performance (Appendix G). We find that these models perform best when evaluated using their training resolution. Therefore, we do not scale the token budget when evaluating CLIP models. This includes instruction-finetuned CLIP models such as Uniir and MagicLens (Wei et al., 2023; Zhang et al., 2024).

| | MSCOCO (5K test set) | | | | | | Flickr30K (1K test set) | | | | | |
| | Image → Text | | | Text → Image | | | Image → Text | | | Text → Image | | |
| Model | R@1 | R@5 | R@10 | R@1 | R@5 | R@10 | R@1 | R@5 | R@10 | R@1 | R@5 | R@10 |
|---|---|---|---|---|---|---|---|---|---|---|---|---|
| CLIP (Radford et al., 2021) | 58.4 | 81.5 | 88.1 | 37.8 | 62.4 | 72.2 | 88.0 | 98.7 | 99.4 | 68.7 | 90.6 | 95.2 |
| ALIGN Jia et al. (2021) | 58.6 | 83.0 | 89.7 | 45.6 | 69.8 | 78.6 | 88.6 | 98.7 | 99.7 | 75.7 | 93.8 | 96.8 |
| FLAVA (Singh et al., 2022) | 42.7 | 76.8 | - | 38.4 | 67.5 | - | 67.7 | 94.0 | - | 65.2 | 89.4 | - |
| FILIP * (Yao et al., 2021) | 61.3 | 84.3 | 90.4 | 45.9 | 70.6 | 79.3 | 89.8 | 99.2 | 99.8 | 75.0 | 93.4 | 96.3 |
| CoCa * (Yu et al., 2022) | 66.3 | 86.2 | 91.8 | 51.2 | 74.2 | 82.0 | 92.5 | **99.5** | **99.9** | **80.4** | **95.7** | **97.7** |
| OpenCLIP-G (Cherti et al., 2023a) | 67.3 | 86.9 | 92.6 | **51.4** | 74.9 | **83.0** | 92.9 | 99.3 | 99.8 | 79.5 | 95.0 | 97.1 |
| EVA-02-CLIP-E+ (Sun et al., 2023) | 68.8 | 87.8 | 92.8 | 51.1 | **75.0** | 82.7 | **93.9** | 99.4 | 99.8 | 78.8 | 94.2 | 96.8 |
| ABC (ours) | **69.2** | **87.9** | **93.2** | 47.6 | 72.1 | 80.6 | 90.7 | 99.0 | 99.5 | 74.6 | 92.6 | 95.45 |

Table 1: Comparison of retrieval performance on MSCOCO Lin et al. (2015) and Flickr30K Plummer et al. (2016) datasets (Karpathy split). Best performance is **bold**, second best is underlined. **\*** indicates a closed-weight model.

| | CLIP | OpenCLIP | SigLIP | UniIR | MagicLens | BLIP2 | MMRet | ABC (ours) |
|---|---|---|---|---|---|---|---|---|
| **Classification (9 tasks)** | | | | | | | | |
| ImageNet-1K Russakovsky et al. (2015) | 55.8 | 63.5 | 45.4 | 58.3 | 48.0 | 10.3 | 49.1 | **71.2** |
| HatefulMemes Kiela et al. (2021) | 51.1 | 51.7 | 47.2 | **56.4** | 49.0 | 49.6 | 51.0 | 52.1 |
| VOC2007 Everingham et al. | 50.7 | 52.4 | 64.3 | 66.2 | 51.6 | 52.1 | 74.6 | **81.4** |
| SUN397 Xiao et al. (2010) | 43.4 | 68.8 | 39.6 | 63.2 | 57.0 | 34.5 | 60.1 | **71.8** |
| Place365 López-Cifuentes et al. (2020) | 28.5 | 37.8 | 20.0 | 36.5 | 31.5 | 21.5 | 35.3 | **40.7** |
| ImageNet-A Djolonga et al. (2020) | 25.5 | 14.2 | 42.6 | 9.8 | 8.0 | 3.2 | 31.6 | **49.4** |
| ImageNet-R Hendrycks et al. (2021) | 75.6 | 83.0 | 75.0 | 66.2 | 70.9 | 39.7 | 66.2 | **86.8** |
| ObjectNet Barbu et al. (2019) | 43.4 | 51.4 | 40.3 | 32.2 | 31.6 | 20.6 | 49.2 | **67.7** |
| Country-211 Radford et al. (2021) | **19.2** | 16.8 | 14.2 | 11.3 | 6.2 | 2.5 | 9.3 | 18.5 |
| *All Classification* | 43.7 | 48.8 | 43.2 | 44.5 | 39.3 | 26.0 | 47.4 | **60.0** |

Table 2: Zero-shot classification results on MMEB (Jiang et al., 2024). We compare with pretrained CLIP models (Radford et al., 2021; Cherti et al., 2023a), instruction finetuned models derived from CLIP (Wei et al., 2023; Zhang et al., 2024) and other LLM backbone approaches (Li et al., 2023; Zhou et al., 2024).

### 4.3 Zero-shot Retrieval and Classification

We first evaluate the quality of **ABC** embeddings, as controlling low-quality visual embeddings is useless. To test embedding quality, we use two zero-shot tasks: retrieval from a pool of candidates and image classification. For retrieval, we use MSCOCO (Lin et al., 2015) and Flickr30K (Plummer et al., 2016). For image classification, we use MMEB Jiang et al. (2024), which provides splits of many standard image classification tasks, such as ImageNet-1K (Russakovsky et al., 2015), VOC-2007 (Everingham et al.), and ObjectNet (Barbu et al., 2019). We note that the classification labels in MMEB are short, often only one or two words. This is problematic for image-captioning models that have been trained on full sentences (Yu et al., 2022). To alleviate this issue, we use Radford et al. (2021)'s technique of embedding classification labels in a sentence template. We use "`A photo of a {label}.`" as our template for all classification evaluations.

In Table 1, our model demonstrates strong image-to-text retrieval capabilities, achieving competitive performance to models that have been contrastively trained with hundreds of GPUs and massive batch sizes (Sun et al., 2023; Cherti et al., 2023a). We achieve the best performance in MSCOCO image-to-text retrieval. Comparatively, our text-to-image retrieval is weaker, as we under-perform both EVA-02-CLIP-E+ and OpenCLIP-G. Averaged across all classification tasks (Table 2), we achieve 11.2% better accuracy than the next-best model. On ImageNet-1K (Russakovsky et al., 2015), **ABC** has 7.7% better zero-shot accuracy than OpenCLIP. **ABC**'s performance on established benchmarks such as MSCOCO and ImageNet-1K indicate that the pretrained model's embeddings are useful. In Section 4.7, we evaluate the fine-tuned adapter's ability to embed images with text instructions into the pretrained model's embedding space.

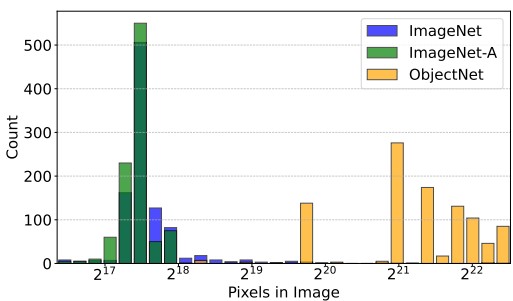
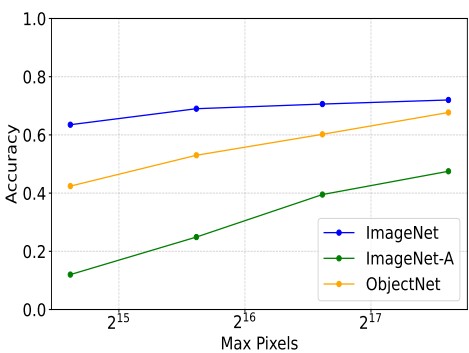

Figure 4: **(Left)** Pixel distributions of benchmarks. ObjectNet (Barbu et al., 2019) contains significantly higer resolution than ImageNet (Russakovsky et al., 2015), which is downscaled. **(Right)** Accuracy on benchmarks when increasing the pixel count (tokens) used by vision encoder when creating embeddings.

### 4.4 Image Resolution and Embedding Quality

Our VLLM backbone, Qwen2-VL-7B (Wang et al., 2024b), supports image inputs with variable resolution. This allows the user to effectively trade-off image resolution with inference speed by adjusting the number of tokens output by the vision encoder. We examine how this trade-off influences embedding quality for image classification. We find that performance on certain tasks, like ObjectNet (Barbu et al., 2019) and ImageNet-A (Djolonga et al., 2020), strongly correlate with the resolution used in the VLLM vision encoder (Figure 4). On average, ObjectNet images have 13 times more pixels than those from ImageNet-1K. When the number of tokens produced by the vision encoder is scaled up, we see a large improvement on ObjectNet, with accuracy increasing by 23.4%. However, lower resolution benchmarks like ImageNet-1K do not benefit nearly as much from scaling resolution. Notably, a smartphone camera takes photos with significantly higher resolution than images in ImageNet-1K or ObjectNet, by default (Apple Inc., 2024). This motivates the need for benchmarks containing larger resolution images. Existing low-resolution benchmarks may underestimate the capabilities of models that can natively utilize high-resolution images. Furthermore, we find that correctly classifying the "natural adversarial examples" of ImageNet-A (Djolonga et al., 2020) is largely a function of resolution. Our accuracy on ImageNet-A increases from only 12% to 47.5% simply by scaling the resolution used by the vision encoder during evaluation.

### 4.5 Mined Negatives and Temperature

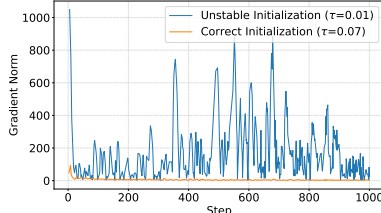 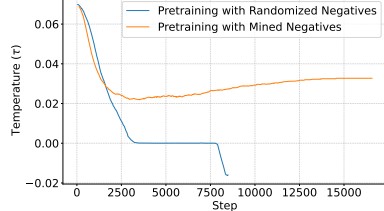 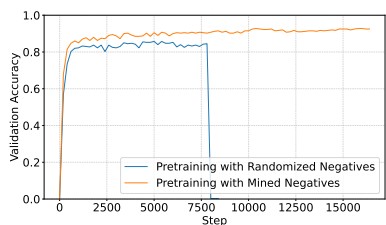

Figure 5: Visualization of temperature ($\tau$) and its effects on training dynamics. **(Left)** The effect of temperature initializations on gradient norm while training our negative mining model. **(Middle)** Temperature behavior during pretraining, without and without mined negatives. **(Right)** Validation accuracy during pretraining, with and without mined negatives.

The setting of the temperature hyperparameter ($\tau$) is known to be crucial for contrastive training (Jia et al., 2021). However, $\tau$ is often treated as a fixed hyperparameter (Jiang et al., 2024; Wei et al., 2023). A large fixed setting of $\tau$ results in a significantly worse model (Wang & Liu, 2021), whereas a too small setting can result in training instability (Figure 5). Therefore, we find that optimizing $\tau$ throughout the pretraining process is crucial. In shorter runs like training our negative mining model, we find that shows that *both* the initialization and optimization of $\tau$ is crucial. For our negative mining model, we find $\tau = 0.07$ to be a temperature initialization that is both stable and performant.

**Mined negatives and temperature.** In Table 3, we ablate over the number of mined hard negatives used during pretraining. Each pretraining run consists of 500 steps with a batch size of 256. Introducing 6 mined hard negatives improves top-1 image to text retrieval by 9.5% and top-1 text to image retrieval by 9.1%. Furthermore, we find that *the mined negatives are essential for training stability.* Figure 5 shows our pretraining run with a batch size of 128 images queries and 1024 text candidates. We compare two runs: one

| # Negatives | Image → Text | Text → Image |
|---|---|---|
| 0 | 50.4 | 35.7 |
| 3 | 58.4 | 44.0 |
| 6 | 59.9 | 44.8 |

Table 3: Top-1 retrieval performance on MSCOCO with varying number of mined hard negatives.

using randomized negatives sampled uniformly randomly from our dataset and the other our mined hard negatives. We find that if randomized negatives are used instead of our mined negatives, the optimizer tends to push $\tau$ very close to 0. This results in numerical instability, loss spikes, and eventually catastrophic failure. When our mined negatives are used in our pretraining, the temperature stabilizes at a very reasonable value of around 0.03.

## 4.6 VLLM Backbone

We explore how the choice of VLLM backbone has an effect on the quality of our multimodal embedding. We ablate over 3 popular choices of VLLM backbone, all at approximately the 8 billion parameter scale: Qwen2-VL-7B (Wang et al., 2024b), our chosen backbone. InternVL-2-8B from the internVL family of VLMs (Chen et al., 2024). Lastly, LLaVA-NeXT (Liu et al., 2024a) with Mistral-7B (Jiang et al., 2023) as its LLM component. We pretrain each model for 1000 steps with a query batch size of 128 and candidate batch size of 1024.

| Model | Accuracy$_{val}$ | MMLU$_{val}$ | MMBench$_{EN}$ |
|---|---|---|---|
| **LLaVA-NeXT** | 62.6 | 35.3 | 68.7 |
| **InternVL-2-8B** | 65.9 | 51.8 | 81.7 |
| **Qwen2-VL-7B** | **70.2** | **54.1** | **83.0** |

Table 4: Ablation over VLLM backbone choice.

Table 4 shows the validation accuracy of **ABC** with different backbones. We find that our backbone choice, Qwen2-VL-7B, produces the best results. We also note each backbone's performance on two standard generative VLLM benchmarks: MMMU Yue et al. (2024) and MMBench Liu et al. (2024b). We find that performance after contrastive training strongly correlates with the performance of the backbone on generative tasks. This indicates that training better VLLMs naturally results in better backbones for our model.

## 4.7 Controlling Embeddings using Natural Language

Table 5 shows **ABC**'s performance on the VQA split of MMEB. In just 100 training steps using instructions, our model surpasses MMRet (Zhou et al., 2024) on VQA, which was trained with millions of instructions. Interestingly, CLIP derivatives fine-tuned with instructions such as MagicLens (Zhang et al., 2024) and UniIR (Wei et al., 2023) do not perform significantly better than their respective baselines, despite MMEB providing instructions for each task. This evidences that encoder-only embedding models struggle to effectively utilize natural language instructions (Jiang et al., 2024).

**Problems with VQA.** A VQA benchmark should require the use of both modalities *together*, and using only the image or the text should be insufficient to accomplish the task. We note that ensuring this property requires inspecting not only the queries but the candidate pool as well. Consider Figure 6, an example from

| | CLIP | OpenCLIP | SigLIP | UniIR | MagicLens | BLIP2 | MMRet | ABC (ours) |
|---|---|---|---|---|---|---|---|---|
| **VQA (10 tasks)** | | | | | | | | |
| OK-VQA Marino et al. (2019) | 7.5 | 11.5 | 2.4 | 25.4 | 12.7 | 8.7 | 28.0 | **48.1** |
| A-OKVQA Schwenk et al. (2022) | 3.8 | 3.3 | 1.5 | 8.8 | 2.9 | 3.2 | 11.6 | **37.3** |
| DocVQA Mathew et al. (2021b) | 4.0 | 5.3 | 4.2 | 6.2 | 3.0 | 2.6 | 12.6 | **28.5** |
| InfographicsVQA Mathew et al. (2021a) | 4.6 | 4.6 | 2.7 | 4.6 | 5.9 | 2.0 | **10.6** | 7.9 |
| ChartQA Masry et al. (2022) | 1.4 | 1.5 | 3.0 | 1.6 | 0.9 | 0.5 | 2.4 | **11.7** |
| Visual7W Zhu et al. (2016) | 4.0 | 2.6 | 1.2 | 14.5 | 2.5 | 1.3 | 9.0 | **25.6** |
| ScienceQA Lu et al. (2022) | 9.4 | 10.2 | 7.9 | 12.8 | 5.2 | 6.8 | 23.3 | **26.3** |
| VizWiz Gurari et al. (2018) | 8.2 | 6.6 | 2.3 | 24.3 | 1.7 | 4.0 | 25.9 | **29.4** |
| GQA Hudson & Manning (2019) | 41.3 | 52.5 | 57.5 | 48.8 | 43.5 | 9.7 | 41.3 | **60.1** |
| TextVQASingh et al. (2019) | 7.0 | 10.9 | 1.0 | 15.1 | 4.6 | 3.3 | 18.9 | **35.4** |
| *All VQA* | 9.1 | 10.9 | 8.4 | 16.2 | 8.3 | 4.2 | 18.4 | **31.0** |

Table 5: Zero-shot VQA results on MMEB (Jiang et al., 2024). Best is **bold**, second best is underlined.

the Visual7W task in MMEB. When the candidate pool is examined, it is clear that there is only one answer that is plausible, given the image. Therefore, the instruction is not required to accomplish the VQA task. This a common pitfall when adapting open-ended generative VQA benchmarks into fixed candidate pool embedding tasks.

**CtrlBench.** We construct `CtrlBench`, an instruction-controlled retrieval benchmark. `CtrlBench` has a similar format to the Flicker30K (Plummer et al., 2016) test split. However, instead of each image having multiple valid captions, we instead provide an instruction from which the model can infer the most relevant text candidate. Therefore, `CtrlBench` tests both retrieval and instruction following capabilities. To construct `CtrlBench`, we sample a 1000 images from ADE20K (Zhou et al., 2017). To create 5000 instructions and text candidate pairs, we generate 5 instructions for each image, each corresponding to a distinct aspect of the image. To accomplish this, we use Gemini 2.5 Flash (Team et al., 2025). We provide the exact prompt that we use in Appendix B. As each image has 5 associated captions in the candidate pool, a model that cannot utilize instructions can (in expectation) achieve at most 20% R@1 on the benchmark. Therefore, `CtrlBench` requires the embedding model to use the provided instructions to disambiguate visual queries. We deduplicate captions from the dataset to prevent ambiguity in our text candidates, resulting in `CtrlBench` being slightly smaller than 1000 images.

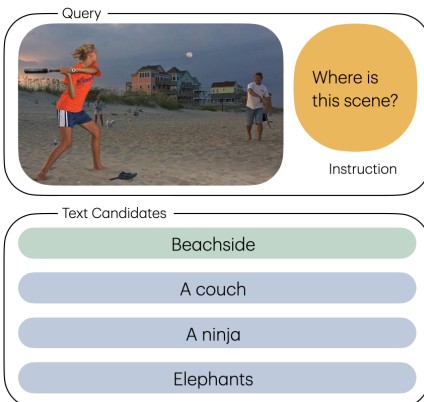

Figure 6: An example of a VQA task from the Visual7W (Zhu et al., 2016) split of MMEB (Jiang et al., 2024).

| Model | R@1 | R@5 | R@10 |
|---|---|---|---|
| **UniIR** Wei et al. (2023) | 0.9 | 2.59 | 4.30 |
| **MagicLens** Zhang et al. (2024) | 17.0 | 37.33 | 48.47 |
| **VLM2Vec** Jiang et al. (2024) | 2.95 | 8.49 | 12.36 |
| **GME-Qwen2VL-2B** Zhang et al. (2025) | 20.92 | 46.33 | 57.27 |
| **GME-Qwen2VL-7B** Zhang et al. (2025) | 24.75 | 54.48 | 65.27 |
| **ABC** (pretrained) | 11.71 | 32.48 | 42.69 |
| **ABC** (finetuned) | **50.94** | **78.43** | **87.47** |

Table 6: Recall@K on `CtrlBench`.

Table 6 shows the performance of multimodal embedding models on `CtrlBench`. We compare with 2 instruction fine-tuned CLIP derivatives: UniIR Wei et al. (2023) and MagicLens Zhang et al. (2024) as well as VLM2Vec Jiang et al. (2024) and GME (Zhang et al., 2025), two concurrent works that adapt VLLMs into multimodal embedding models. We find that neither of the CLIP-based architectures have above 20% R@1 on `CtrlBench`, the performance that indicates that the model is non-trivially utilizing instructions. We find that VLLM architectures are better at instruction-controlled retrieval. **ABC** and VLM2Vec have R@1 above 20%, proving that they can non-trivially utilize instructions.

## 5 Discussion and Related Work

### 5.1 Distribution Overlap in Training and Evaluation

Throughout our experiments, we found that several training data sources overlap with commonly used benchmarks. Several benchmark are constructed from the same underlying data source as training data or directly derived from training data (Barbu et al., 2019; Russakovsky et al., 2015; Plummer et al., 2016; Lin et al., 2015). This makes evaluating "zero-shot" capabilities difficult. To the author's knowledge, our pretraining data (Sharma et al., 2018) and instruction-finetuning data Krishna et al. (2016) does not overlap with any benchmarks we evaluate against, and all evaluations (Tables 1, 2, 5 and 6) are out-of-distribution.

### 5.2 Decoupling Pretraining and Instruction Fine-tuning

We find that adapting an instruction fine-tuned VLLM to use instructions is relatively cheap. Our instruction fine-tuning completes in less than an hour on a couple A100 GPUs. This allowed us to quickly iterate on our instruction fine-tuning stage, without pretraining from scratch. Comparatively, training a VLLM to create high-quality embeddings is much more resource intensive. Our pretraining run barely fits into the 640 GB of VRAM provided by a single A100 node, and takes several days to complete. Furthermore, our work indicates that these models could benefit substantially from further scaling (Appendix D).

### 5.3 Alternative Approaches

MagicLens (Zhang et al., 2024) and UniIR (Wei et al., 2023) are multimodal embedding models that fuse CLIP embeddings. By combining multiple CLIP models, they take image-text pairs and align them with image-text pairs using contrastive loss and instruction finetuning. Due to late fusion of modalities, these architectures struggle to use instructions to modify representations (Jiang et al., 2024). Recently, VLM2Vec Jiang et al. (2024) and GME (Zhang et al., 2025) have also adopted contrastive training on VLLM backbones to produce multimodal embeddings. However, these approaches make extensive use of human-annotated instruction datasets, with hundreds of thousands of labeled examples used during training. This limits the scalability and adaptability of their method.

### 5.4 Important Factors during Pretraining

Prior work has largely focused on what architectural adaptations to the make the VLLM to convert it into an embedding model. These include the attention mask, how the embedding is pooled and adapter architecture (BehnamGhader et al., 2024; Lee et al., 2024; Jiang et al., 2024). Throughout our experiments, we find that most of these choices are often interchangeable or only produce marginal improvements (Appendix C). However, we find that many of the crucial factors when training CLIP models are also important when adapting VLMs. In particular, well-chosen data, batch size and number of samples seen during training are all important factors (Appendix D), just like with CLIP models (Gadre et al., 2023; Cherti et al., 2023a).

## 6 Conclusion

We introduce **ABC**, a multimodal embedding model that leverages a VLLM backbone to control image representations via natural language instructions. It achieves strong zero-shot results on a variety of multimodal tasks, spanning retrieval, classification, and VQA. **ABC** consists of a multi-stage training process, which isolates the computationally expensive contrastive pretraining from a lightweight instruction finetuning phase. We explore what factors are the most crucial when adapting VLMs to output multimodal embeddings. In particular, we find that vision encoder resolution, contrastive loss temperature and VLLM backbone choice are all important factors. Lastly, we design `CtrlBench` to measure our model's ability to use instructions to accomplish subtle natural language-guided retrieval tasks.

## 7 Broader Impact Statement

The ability to exhibit greater control over visual representations, especially those used information retrieval systems, has potential for both positive and negative impacts. On the positive side, it allows users greater control over visual information retrieval. This capability is particularly useful for visually-impaired users that can struggle to directly use and interpret images. However, introducing natural language control over visual representations has potentially negative impacts. We highlight two potential harms: (1) Accidental introduction of bias due to natural language conditioning. Large language models exhibit similar biases to humans (Guo et al., 2024). LLM representations used in information retrieval may adopt these biases. Furthermore, allowing users natural language control over visual representations may result in users accidentally introducing their own biases into visual retrieval. (2) Malicious control over information retrieval. Introducing natural language control over visual representations creates an entry-point to filter or modify

what information is retrieved by downstream users. This is alarming in the internet age, where control over information access is a powerful tool for authoritarian control.

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

## A Instruction Fine-tuning Prompt Design

We use the following prompt template when using GPT-4o to generate instructions:

```
"<Image> Given this image, provide a user prompt where the following caption would be a
reasonable answer:  {Caption}.  Only return the prompt."
```

Where `{Caption}` is the caption for the bounding box from Visual Genome. We use a temperature of 0 (deterministic) when generating instructions. We choose this template to collect a diverse set of instructions that a user would plausibly ask.

## B `CtrlBench` Prompt Design

We use the following template in Gemini 2.5 Flash for each of the 1000 images:

```
"<Image> Your task is to identify and describe 5 distinct objects or elements in this image. For
    each object provide a short sentence description of it. Please be specific in your
    description. Then, for each description, provide a user prompt where that description caption
    would be a reasonable answer. Output your answer as JSON in the following template: {"desc_1":
    <DESCRIPTION 1>, "prompt_1": <PROMPT 1>, "desc_2": <DESCRIPTION 2>, "prompt_2": <PROMPT 2>,
    "desc_3": <DESCRIPTION 3>, "prompt_3": <PROMPT 3>, "desc_4": <DESCRIPTION 4>, "prompt_4":
    <PROMPT 4>, "desc_5": <DESCRIPTION 5>, "prompt_5": <PROMPT 5>}
```

Save on API costs, we generate all the candidates and queries for each in a single prompt.

## C Architecture Ablation

LLM2Vec (BehnamGhader et al., 2024) introduced several architecture modifications for adapting LLMs into text embedding models. In the multimodal setting, we find that using casual vs. bidirectional attention is marginal for improving model accuracy (Figure 7).

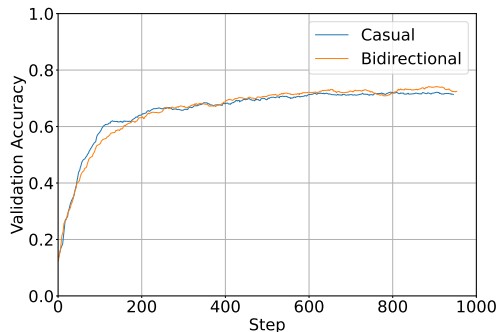 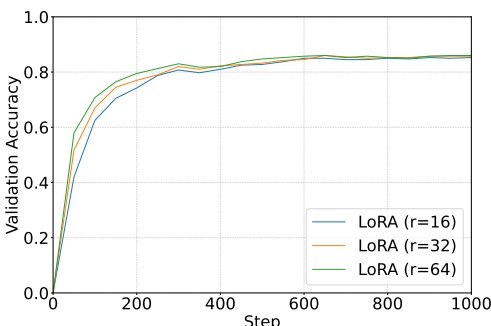

Figure 7: **(Left)** Causal vs Bidirectional attention. **(Right)** LoRA adapter rank ablation.

On average, bidirectional attention *slightly* outperforms casual attention. Ablating over adapter rank, we find that different rank adapters tend to converge to the same accuracy, with higher rank adapters performing a fraction of a percentage better. Overall, better data and scaling (Appendix D) present a much more promising direction for improving embedding quality.

## D Scaling Training

We find that batch size and number of samples seen during training are both important factors. By increasing our batch size by 4x, from 128 queries and 1024 candidates to our full pretraining run size (512 queries and 4096 candidates), our validation accuracy on an 800 sample validation batch using in-batch negatives

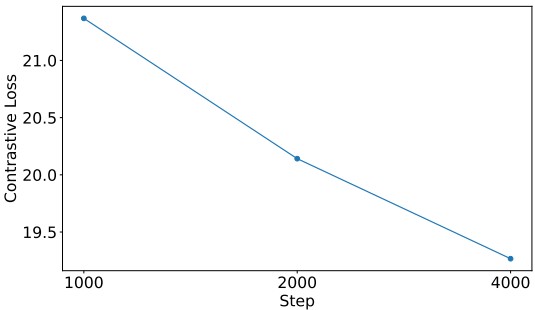

Figure 8: More samples seen vs. loss.

increases from 92.5% to 95.0%. We control for total samples seen by training the smaller batch size model for 4x more steps. Interestingly, this effect is even more pronounced on OOD validation data. For example, on a similar batch of MSCOCO data, accuracy increases from 73.8% to 84.0%. Therefore, techniques for scaling batch size under limited VRAM, such as GradCache Gao et al. (2021), are a promising direction to improve our pretrained model. Furthermore, we find that more steps (more samples seen) is also a straightforward way to increase the pretrained model's performance. As shown by Figure 8, doubling step count steadily decreases loss during our pretraining run.

# E  Comparison to VLM2Vec

With the MMEB benchmark, Jiang et al. (2024) also include a multimodal embedding model: VLM2Vec. We exclude VLM2Vec from Tables 1, 2 and 5 as it has been trained on MSCOCO and several tasks from the MMEB benchmark. We provide the comparison of the two models here while specifying which tasks are out of distribution (OOD) for VLM2Vec. All tasks are OOD for our model.

|  | VLM2Vec | ABC (ours) |
| --- | --- | --- |
| ImageNet-1K (Russakovsky et al., 2015) | 65.6 | **71.2** |
| HatefulMemes Kiela et al. (2021) | **67.1** | 52.1 |
| VOC2007 Everingham et al. | **88.6** | 81.4 |
| SUN397 Xiao et al. (2010) | **72.7** | 71.8 |
| Place365 López-Cifuentes et al. (2020) | **42.6** | 40.7 |
| ImageNet-A (Djolonga et al., 2020) | 19.3 | **49.4** |
| ImageNet-R (Hendrycks et al., 2021) | 70.2 | **86.8** |
| ObjectNet Barbu et al. (2019) | 29.5 | **67.7** |
| Country-211 Radford et al. (2021) | 13.0 | **18.5** |
| *Average* | 52.1 | **60.0** |
| *Average OOD* | 34.9 | **52.6** |

Table 7: Classification results for VLM2Vec and **ABC (ours)**. Gray background indicates that VLM2Vec has been trained on this task.

Table 7 compares the performance of our model to VLM2Vec Jiang et al. (2024). Gray rows indicate tasks that VLM2Vec is trained on. On average, VLM2Vec performs better on tasks that are in its distribution, while our model outperforms on tasks where both models are OOD. Interestingly, our model performs better on ImageNet-1K even though it is not trained on it.

## F    Results on VLLMs without Contrastive Training

Can dense embedding be extracted from VLLM's without additional contrastive training? We experiment with two common methods for extracting dense embeddings from last-layer hiddens: **(1)** Using the last token in the sequence (Jiang et al., 2024). **(2)** Averaging over the sequence dimension (BehnamGhader et al., 2024).

| Pooling Method | Direction | Top-1 | Top-5 | Top-10 |
|---|---|---|---|---|
| Mean Pooling | i2t | 6.30% | 16.18% | 23.70% |
| | t2i | 3.41% | 9.25% | 13.39% |
| Last Token | i2t | 0.06% | 0.26% | 0.36% |
| | t2i | 0.08% | 0.30% | 0.73% |

Table 8: Comparison of pooling methods across retrieval directions and on MSCOCO.

On MSCOCO retrieval we find that mean pooling significantly outperforms using the last token hidden (Table 8). However, neither method performs very strong. Both methods are significantly worse than a zero-shot CLIP model on MSCOCO (Table 1).

## G    Inference Settings for CLIP

| Resolution | Image → Text | Text → Image |
|---|---|---|
| 224x224 | 64.94 | 48.02 |
| 336x336 | 63.28 | 47.42 |
| 448x448 | 61.98 | 45.97 |

Table 9: OpenClip-ViT-g-14 accuracy (top-1) on MSCOCO at different image resolutions. Unlike **ABC**, accuracy degrades when scaling token budgets beyond OpenClip-ViT-g-14's native $224 \times 224$ resolution.

We find that CLIP models perform best at their pretraining resolution. As shown in Table 9 accuracy consistently degrades when scaling the token counts of CLIP models beyond their native resolution. It would be unfair to evaluate models at a setting that is both more compute intensive and less performant. Therefore, we evaluate CLIP models at their native resolution and token count across all experiments.

## H  CtrlBench Examples

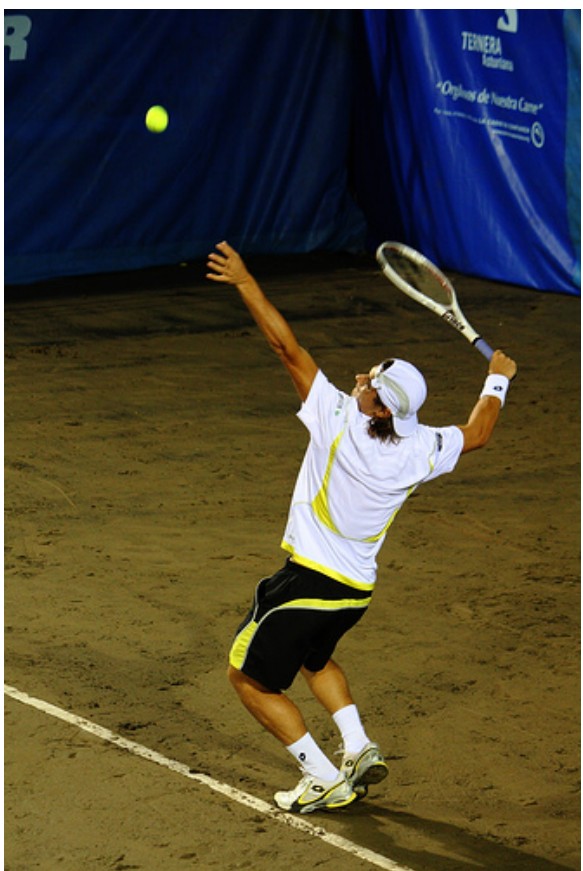

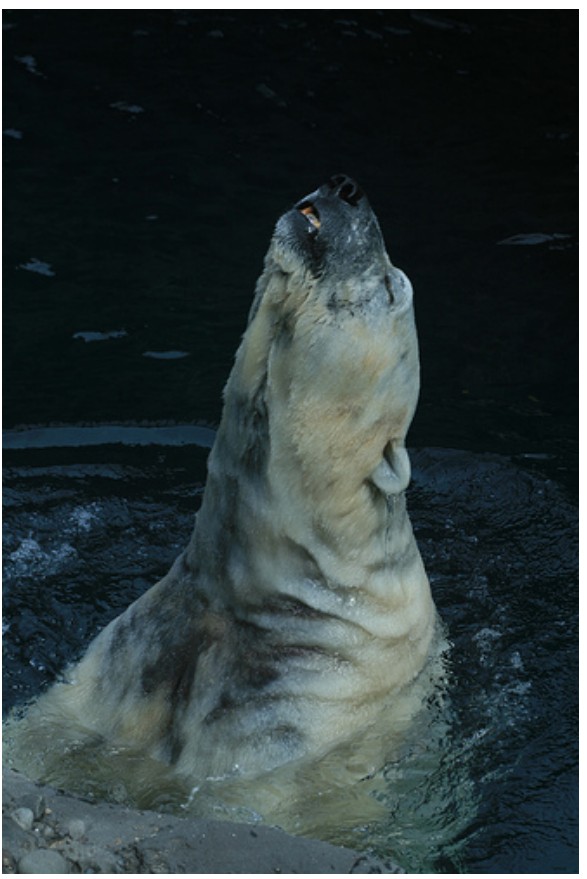

**Q**: What is happening in the top left corner of the image?
**A**: Tennis ball in the air

**Q**: What is the player about to hit with his racket?
**A**: A round tennis ball

**Q**: What color are the shorts the tennis player is wearing?
**A**: black and yellow shorts

**Q**: What action is taking place on the tennis court in the image?
**A**: A man serving a tennis ball.

**Q**: What's prominently dividing the ground area in the image?
**A**: White line on a court

**Q**: What part of the animal is prominently displayed in the image?
**A**: the neck of a polar bear

**Q**: What colors can you see on the bear in the image?
**A**: black, brown and white bear neck

**Q**: What stands out to you about the bear's appearance in this image?
**A**: the bear has such tiny ears for such a huge animal

**Q**: What is the bear doing with its eyes in this picture?
**A**: the bear has his eyes closed

**Q**: What's the polar bear swimming in?
**A**: a body of water

