# OpenReview forum: "ABC: Achieving Better Control of Visual Embeddings using VLLMs"
_TMLR — Accepted by TMLR_

### Review · Reviewer_JnhS · 2025-06-08

**Summary Of Contributions:**

Achieve Better Control (ABC) is a visual language model (VLM) where the authors propose a method to let users steer image embeddings via contextual language prompts. As opposed to CLIP-style embeddings which measures alignment between language and image in a static context, ABC uses low-rank adapters to control image embeddings based on queries. The high-level goal seems to be better adaptation in VLMs, and in practical terms the authors focus on VQA and retrieval tasks.

The authors astutely point out issues with existing VQA benchmarks, such as cases where queries can be answered more or less independently from visual data, i.e. $p(a \mid q, i) = p(a \mid q)$, where $(q,a)$ is a query-answer pair and $i$ is an image embedding. In response, the authors craft a new benchmark that address these issues directly. The authors motivate conditional "control" via ambiguous VQA triplets, that need disambiguation to be meaningfully parsed, but does not explicitly formalise what is meant by ambiguity, nor how this is solved by the proposed method.

The authors conclude by stating that their approach provides strong zero-shot results in downstream tasks, and emphasise their contributions in evaluating the effect of resolution, temperature scaling, and backbone; all considered important for these tasks. However, some details regarding the effect of resolution is left muddled, as we mention in W2.

**Audience:**

No

**Broader Impact Concerns:**

The line of research proposed by the authors can arguably permit more aggressive control of information. Since ABC lets the querier push an instruction through a LoRA that perturbs the image representation before any downstream scoring, whoever controls the prompt also controls what “counts” as similarity. *This clearly warrants broader impact statement on the work.*

The capability for controlling image embeddings can of course go both ways. Taking a positive example; a radiologist can focus retrieval on *tubular opacities* rather than overall lung texture to more succinctly steer queries, contributing to better healthcare for patients. However, the very same approach can be used to suppress disfavoured political views or human rights violations. Suppose an authoritarian agent attaches the instruction *“ignore any protest banners”* to every incoming street photo before it is indexed. Any retrieval pipelines that trust ABC embeddings (as most of society would in interacting with VLMs) *would quietly forget visuals of dissent* even though the visual data clearly would suggest otherwise.

The issue is exacerbated by the fact that the approach provide tiny, hot-swappable changes to models, altering the outputs of what could be perceived by users as the same model. Hence, outputs from a trusted model can be subtly changed to provide targeted opinions designed to alter a users perceptions of the world with little side effects.

In other words, this reviewer points out that the ethical implications of the work is largely ignored in the current draft of the manuscript, and clearly needs addressing before publication.

**Claims And Evidence:**

No

**Requested Changes:**

- This reviewer has issues on the effect of controlling image embeddings to improve VLM tasks. Can the authors successfully demonstrate that the results are indeed a factor of the conditioning, and not confounding architectural modifications and training regimes? Demonstrating this is crucial, as the method is motivated by control as a driver, necessitating a controlled experiment.
- Can the authors elaborate and clearly show what type of ambiguity is resolved by conditioning image embeddings on queries? Specifically, better examples are required to delineate this as a clear problem statement. Figure 6 is clearly motivated but does not address ambiguity; rather it demonstrates examples where there is *less* rather than *more* ambiguity in the VQA triplet. A formalisation of what ambiguity means should be included.
- Can the authors provide concrete evidence that ABC eliminates confounders, and control for token budgets in an apples-to-apples comparison? As it stands, there are seemingly many uncontrolled factors that prevents the reader from delineating the contributions of the method from others. Simply increasing token budgets for the proposed method without symmetrically controlling for this in the baselines is not very convincing.
- Similarly, with regard to the the issues mentioned in W2, can the authors address the asymmetry and show that the reported results are not an effect of the experimental design, but rather a property that comes from their proposed method?
- Can the authors show that visual memorisation has not meaningfully contributed to biasing `CtrlBench`? As it stands, the benchmark could have data leakage, and the authors need to convincingly show that this is not an issue for adoption by the community.
- The work is missing a broader impact statement on ethical concerns, as mentioned in the dedicated section. This will need to be addressed.
- As more minor adjustments, details on some aspects on the overall method should be tightened; see W4 for suggestions.

**Strengths And Weaknesses:**

In the assessment of the manuscript, this reviewer has emphasised soundness, motivation, and interesting contributions that can positively impact the research community on a broader scale. By this, we mean that a method can stand out as promising without showing gains in every conceivable downstream task. Additionally, this reviewer has evaluated the work on the clarity, precision, and reproducibility. Based on these criteria, we highlight strengths and weaknesses in the submission.

### Strengths

1. The authors directly address issues in current benchmarks by releasing `CtrlBench`, which looks to alleviate issues where answers can be constructed without even a cursory glance at the source image. Pointing out these types of irregularities in how models are benchmarked and results are reported is always a worthwhile contribution.

2. The paper has several technical details, providing optimisation and contributions in ML-engineering. The authors should be applauded for their commitment to sharing their insights in VLM modelling; which is unfortunately becoming rarer in the field.
    - The proposed two-stage recipe via large-scale contrastive pre-training, followed by 100-step LoRA adapter is a handy setup.
    - The paper is generous in spelling out several optimiser hyper-parameters, temperature schedules and memory management.

3. The idea of conditioning with (as this reviewer understand the approach) inference-time switchable adapters comes across as an interesting approach, short of being entirely novel (related ideas have been explored in other generative modelling approaches). Nevertheless, it is an architectural design component that can be applied more generally as a conditioning step.


### Weaknesses

#### Major Concern: Motivating Ambiguity

The paper comes across as opaque about is purported problem statement. Except for a vague mention of *ambiguity* of image-text-text triplets in the introduction, the paper practically focuses on VQA, zero-shot retrieval and classification in a general setting. This would be fine if the work did not base the motivated approach contingently on what they state as *ambiguity*. Figure 1 is designed to highlight this ambiguity inherent in data, but seemingly fails to demonstrate what type of ambiguity is present in the figure. Why are these ambiguous? The fact that a *scene can provide multiple unrelated factoids* does not necessitate *ambiguity*. Aside from this brief mention, this stated issue on ambiguity is not elaborated on further in the text, and the authors move on to motivate that their method purportedly "solves" this ambiguity problem by modifying image embeddings with adapters. The next mention of ambiguity is presented alongside an example (Figure 6) which clearly shows how a question may be answered independently of the image. This shows a case where $p(a \mid q, i) = p(a \mid q)$, such that the correct answer can be deduced via the query alone. However, *this is wholly unrelated with embedding images conditioned on the query*, which the authors propose as a method to resolve ambiguity. **In short:** *Without formalising what the authors mean by ambiguity, the reader cannot be expected meaningfully ascertain that the motivation and method has any impact on the problem at hand. Increments in results may easily come from other factors, such as negative mining and multi-stage training pipelines*.

#### Major Concern: Assumptions

A second major concern strikes at the epistemological assumptions in the current work. ABC allows for stronger conditioning on visual prompts to tackle ambiguity (whatever the authors decide this may be). While this reviewer would agree that specification in queries can lead to better responses, this can go both ways. If a query has a well-defined response, adapting the visual representation to modify how the image is interpreted could conversely lead to *negative bias and subjectiveness that may be unwanted*, just as it may help to disambiguate. Taking a more classical view, ambiguity can be better resolved by being able to attach uncertainty to responses, and detect nonsensical or unclear queries in multimodal settings. These skills are woefully underrepresented in language models already, and it is not clear that adding more "control" of how the model  interprets an image is actually useful, outside of the field of alignment. **Respectfully**, *this reviewer maintain doubts about the central assumptions motivating the work*.

#### Specific weaknesses

1. The resolution story comes across as muddled. The authors present a gain on ObjectNet when they raise the token budget, then concede that the baseline datasets are down-scaled and “may underestimate the capabilities” of high-res models. If this reviewer is reading this correctly, all ObjectNet numbers are reported with up-scaled inputs, but the baselines stick to 224² crops. This seems like an apples-to-oranges comparison without control for confounding effects. Re-running OpenCLIP-G with similar token budgets would likely close much of the gap.

2. The outlined retrieval in section 4.2 is asymmetric by construction. Image $\rightarrow$ text negatives are mined, text $\rightarrow$ image are not, leading to the very gap the authors report, but in the work this is treated as an “expected” outcome rather than a design flaw to fix.

3. While the authors admit Visual Genome is both the fine-tuning and benchmark source, this is handwaved by a claim that Gemini paraphrasing fixes overlap. Paraphrasing captions does not eliminate visual memorisation. This reviewer worries that the presented methodology might lead to data leakage.

4. The paper contains exposition that is non-formal and unclear, making reproducibility difficult;
    - "We create synthetic instructions that correspond to different aspects of the same image and use these as negatives." It is not clear what is actually done by the authors here. Is this related to Section 3.3 and `CrtlBench`?
    - Where exactly do the LoRA hooks live? An “instruction adapter” is added after the frozen dual-encoder has finished contrastive pre-training, but the authors never state which blocks receive the rank-4 (or 8? 16?) matrices, whether they act on the vision trunk, the projection head, or the joint fusion module, and whether scaling $\alpha$ is fixed or learned.
    - Authors provide an illustration or a prompt string `<Image> Instruction: …` but do not spell out the actual special-token IDs, the segment-type embeddings, or the positional conventions. Does ABC prepend the image CLS before every instruction, or is the vision vector written into a reserved unused token? While these are reporting issues shared with other works in the field (particularly for VLMs), these details matter for the reader.
    - The contrastive pre-training stage is unclear. What are the augmentations applied to the vision side, and what is the exact objective for mined versus random negatives?

5. Aside from the purported benefits in the approach, there are ethical ramifications in allowing users interact with a model that can control image embedding in relation to the query. These are not addressed at all in the current work.

---

> ### Author Response · Authors · 2025-06-24
> **Thank you for your comments! (part 1/2)**
>
> We thank the reviewer for their detailed review and feedback. Please find our responses below.
>
> >Can the authors elaborate and clearly show what type of ambiguity is resolved by conditioning image embeddings on queries? Specifically, better examples are required to delineate this as a clear problem statement. Figure 6 is clearly motivated but does not address ambiguity; rather it demonstrates examples where there is less rather than more ambiguity in the VQA triplet. A formalisation of what ambiguity means should be included.
>
> We agree that a formalization of ambiguity is required, and will update the manuscript to include a background section addressing this topic. Adapting the notion of ambiguity from [1], we define an image-only retrieval/classification problem as *ambiguous* if the candidate pool contains multiple correct (but conceptually distinct) candidates. We will update the introduction and Figure 1 with an example of this problem from CtrlBench. We agree that Figure 6 does not relate to ambiguity, rather it is demonstrating a common construction flaw in VQA tasks. We will revise the manuscript to clarify that the motivation behind CtrlBench is to test retrieval capabilities under visually ambiguous settings **and** mitigate image-only trivialization.
>
> >Can the authors provide concrete evidence that ABC eliminates confounders, and control for token budgets in an apples-to-apples comparison? As it stands, there are seemingly many uncontrolled factors that prevents the reader from delineating the contributions of the method from others. Simply increasing token budgets for the proposed method without symmetrically controlling for this in the baselines is not very convincing.
>
> Across all experiments, we seek to evaluate baselines with their ideal inference settings. However, we find that this is not always aligned with settings that maximize compute. In particular, *CLIP models consistently degrade when scaling tokens budgets beyond their pretraining resolution.* See MSCOCO t2i and i2t results for OpenClip-ViT-g-14 below:
>
> | Resolution | i2t Top-1 | t2i Top-1 |
> |------------|-----------|-----------|
> | 224x224    | 64.94    | 48.02    |
> | 336x336    | 63.28    | 47.42    |
> | 448x448    | 61.98    | 45.97    |
>
> Therefore, we constrain the token budgets of CLIP models to performance optimal settings during evaluations. When evaluating baselines designed to have adaptive token budgets (such as VLM2Vec [2] and GME [3]), we scale token budgets to match our method, accordingly. We will revise Section 4 to discuss our choice of inference settings for baselines and include the above result.
>
> >Similarly, with regard to the the issues mentioned in W2, can the authors address the asymmetry and show that the reported results are not an effect of the experimental design, but rather a property that comes from their proposed method?
>
> The authors agree that "expected result" is poor phrasing and will revise the manuscript to describe our weaker t2i performance as a limitation. Furthermore, we agree that the emergence of this performance gap is interesting and motivates additional investigation. We ablate over the number of mined negatives used (each run consists of 500 steps with a batch size of 256, we report results on MSCOCO) and the bidirectional formulation of contrastive loss: $(\text{i2t} + \text{t2i}) / 2$ adopted by several baselines.
>
> | # Negatives | i2t - i2t loss   | t2i - i2t loss   | i2t - bidirectional loss| t2i - bidirectional loss|
> |-------------|------------------|------------------|------------------------|------------------------|
> | 0           | 50.42            | 35.71            | 47.82                  | 36.79                  |
> | 3           | 58.42            | 43.97            | 56.86                  | 43.29                  |
> | 6           | 59.94            | 44.84            | 58.06                  | 44.09                  |
>
> We find that our negative mining regime increases performance in both i2t and t2i performance. We find that while bidirectional loss improves i2t performance when no text negatives are used, it performs worse than standard i2t loss when text negatives are introduced. Overall, if negative mining is not used, bidirectional loss results in slightly better i2t retrieval performance.
>
> [1] Stelmakh et al., "ASQA: Factoid Questions Meet Long-Form Answers." Proceedings of the 2022 Conference on Empirical Methods in Natural Language Processing, 2022.
> [2] Jiang et al., "VLM2Vec: Training Vision-Language Models for Massive Multimodal Embedding Tasks." The Thirteenth International Conference on Learning Representations, 2025.
> [3] Zhang et al., "GME: Improving Universal Multimodal Retrieval by Multimodal LLMs." arXiv preprint, 2024.

---

> > ### Author Response · Authors · 2025-06-24
> > **Thank you for your comments! (part 2/2)**
> >
> > >What is the exact objective for mined versus random negatives?
> >
> > The training objective is the same, they only differ in how they are sampled. A "randomized" negative is a candidate chosen uniformly randomly from our dataset (excluding the current batch) instead of from our mined negatives candidates (section 3.2). We do this to control for batch size when ablating over how mined negatives effect training dynamics. We will clarify our approach in section 4.4.
> >
> > >Can the authors show that visual memorisation has not meaningfully contributed to biasing CtrlBench? As it stands, the benchmark could have data leakage, and the authors need to convincingly show that this is not an issue for adoption by the community.
> >
> > We agree that data leakage is a significant issue. We are in the process of creating a version of CtrlBench that uses the Open Images Dataset [4] for its visual distribution. We will update the reviewer with a revised version of table 5 when we complete the construction.
> >
> > >This reviewer has issues on the effect of controlling image embeddings to improve VLM tasks. Can the authors successfully demonstrate that the results are indeed a factor of the conditioning, and not confounding architectural modifications and training regimes? Demonstrating this is crucial, as the method is motivated by control as a driver, necessitating a controlled experiment.
> >
> > We update table 5 (evaluation on CtrlBench) to include baseline of ABC without the adapter, which shows ABC without text conditioning:
> >
> > | Model           | R@1   | R@5   | R@10  |
> > |-----------------|-------|-------|-------|
> > | UniIR           | 0.0   | 0.0   | 0.1   |
> > | MagicLens       | 11.7  | 28.4  | 38.5  |
> > | VLM2Vec         | 26.44 | 56.5  | 68.9  |
> > | GME-Qwen2VL-2B  | 9.4   | 24.6  | 33.9  |
> > | GME-Qwen2VL-7B  | 10.7  | 27.6  | 36.9  |
> > | **ABC (pretrained)**  | **5.73** | **15.6** | **22.53** |
> > | **ABC (finetuned)**  | **41.0** | **56.5** | **80.0** |
> >
> > On visual memorization in table 5, we note that our baselines are trained on the *same visual distribution*. Our model is trained on 12,800 images from Visual Genome, which is derived from MSCOCO. The baselines we compare against are trained on hundreds of thousands of images from MSCOCO [2,3,5]. Therefore, although our method may benefit from visual memorization, we expect it benefits much less than the baselines we compare against. Nevertheless, we agree that providing the community with tasks from entirely disjoint visual distributions is important. The added pretrained baseline and Open Images version of CtrlBench will show how much performance results from the text conditioned adapter while controlling for memorization in the visual distribution.
> >
> > >The work is missing a broader impact statement on ethical concerns, as mentioned in the dedicated section. This will need to be addressed.
> >
> > We will update our manuscript to include a broader impact statement that addresses the potential positive (assistance technology for visually impaired users) and negative (accidental introduction of bias through text conditioning, control over information retrieval) impacts of our work.
> >
> > >As more minor adjustments, details on some aspects on the overall method should be tightened; see W4 for suggestions.
> >
> > Thank you for helping us improve the clarity of our method. We will update our manuscript to specify the requested details:
> >
> > 1. A single batch contains the same image multiple times, conditioned on different text. This helps the adapter to learn how to use text-conditioning to modify the image representation. In essence, it is negative mining for text-conditioning.
> >
> > 2. The LoRA hooks of the instruction adaptor are attached to the MLP and $\mathbf{W}^Q,\mathbf{W}^K,\mathbf{W}^V$ projections in each decoder block of the LLM backbone. We use a rank of 16. We set $\alpha$ to be twice the rank during pretraining and fine-tuning.
> >
> > 3. We adopt the templating format from our pretrained backbone:
> > <|vision_start|> <|visual_token|>... <|visual_token|> <|vision_end|> <|im_start|> Instruction: {text} <|im_end|>
> > Where <|vision_start|>, <|vision_end|>, <|im_start|> and <|im_end|> are vocabulary tokens used to distinguish vision and text tokens. If no instruction is provided the last token is <|vision_end|>.
> >
> > 4. Our image processing consists of resizing and normalization. We do not use any cropping. During training we use 256 visual tokens per image.
> >
> > Thank you once again for your detailed feedback. Please let us know if our changes have assuaged your concerns or if you have any additional questions.
> >
> > [4] Kuznetsova et al., "The Open Images Dataset V4: Unified image classification, object detection, and visual relationship detection at scale." arXiv preprint, 2018.
> > [5] Wei et al., “UniIR: Training and Benchmarking Universal Multimodal Information Retrievers.” in Computer Vision - ECCV 2024, vol. 15145, 2024.

---

> > > ### Comment · Reviewer_JnhS · 2025-06-30
> > > **Response to Authors**
> > >
> > > Many thanks to the authors for a detailed comprehensive response.
> > >
> > > ### Motivation & ambiguity
> > >
> > > The authors have proposed a new example of ambiguity in figure 1, and added text to formalise a definition. The ambiguity showcased in the example, where an image has three potential answers for which two seem technically correct. Again, the issue is that this ambiguity is posed to motivate the central idea of the paper, which seems to be how image embeddings need to be conditioned on text queries to disambiguate a VQA triplet. Again, this new example seems to motivate the existence of label noise in VQA tasks rather than the method? As such, the central argument we proposed still stands; *this motivates multilabel approaches to classifying the supplied answer, not conditioning image embeddings on queries*.
> > >
> > > Moreover, even if CtrlBench truly contains image–text pairs with several valid answers, the ability to *steer* an embedding with a textual adapter does **not** guarantee that *all* valid facets are preserved; it only guarantees that **one** facet can be emphasised.  As it stands, this reviewer still maintains that the rebuttal does not completely resolve the issue of what is actually meant by ambiguity.
> > >
> > > As it stands, *the point remains the most clear issue with the approach*, as it is posed as a central theme of investigation in the work.
> > >
> > > ### Pipeline clarity & reproducibility
> > >
> > > The authors do provide new details (rank = 16 LoRA on Q/K/V & MLP, 256 vision tokens, template string).  Some minor details are still somewhat unclear; e.g. size of in-batch candidate pools for negative mining. One potential inconsistency that the author could perhaps resolve; the adapter is described as being added “after the frozen dual encoder”, yet the hooks are placed *inside every decoder block* of the LLM backbone?
> > >
> > > On the whole, **the authors deserve credit for going above and beyond the standard reporting in VQA / VLM tasks in outlining their methodology**; as mentioned in the original review.
> > >
> > >
> > > ### Asymmetric retrieval & mined negatives
> > >
> > > The authors provide an ablation table for a 500 step run, however the real model is quite a bit larger. More importantly, even if the property that "bidirectional loss is worse" even under scale holds, it seems that the authors do not demonstrate text-to-image results when **both** directions receive mined negatives. This is what we meant by asymmetry in our original comment. There is still some doubt whether the performance boost is an artefact of the experimental design.
> > >
> > > ### Resolution / token budget comparison
> > >
> > > The new CLIP degradation table suggests 224-token CLIP is optimal, but it is still a little unclear how this was performed. Does that experiment rescale the *entire* image, rather than **patch size vs. patch count** trade-off that ABC exploits? If so, there is a worry that there may still be a discrepancy between baselines and the proposed model.
> > >
> > > ### CtrlBench leakage
> > >
> > > The authors grant that leakage is a concern, and propose switching to OpenImages to source the dataset. However, it is not clear on what the authors mean when they say they will update reviewers on this task upon completion? Does this involve a complete restructuring on `CtrlBench`? If this is to be taken as a central contribution, re-sourcing the dataset from scratch during rebuttal seems unlikely to be achievable.
> > >
> > > The update on table 5, the evaluation on `CtrlBench` is now including a helpful row “ABC-pretrained". However, it is not clear that this is on the "old" `CtrlBench`? The performance jump is very significant, and data leakage remains a concern for this reviewer. As of yet, this seems to be very much up in the air.
> > >
> > > ### Ethics / broader-impact
> > >
> > > We are pleased to see that the authors added a broader impact statement, addressing the concerns of the reviewer.

---

> > > > ### Author Response · Authors · 2025-07-03
> > > > **Thank you for your response! (1/2)**
> > > >
> > > > Thank you very much for your detailed replies and time. Your feedback has been invaluable for improving our work. Please see our responses below, we will upload a corresponding revised manuscript promptly.
> > > >
> > > > >The authors have proposed a new example of ambiguity in figure 1, and added text to formalise a definition. The ambiguity showcased in the example, where an image has three potential answers for which two seem technically correct. Again, the issue is that this ambiguity is posed to motivate the central idea of the paper, which seems to be how image embeddings need to be conditioned on text queries to disambiguate a VQA triplet. Again, this new example seems to motivate the existence of label noise in VQA tasks rather than the method? As such, the central argument we proposed still stands; this motivates multilabel approaches to classifying the supplied answer, not conditioning image embeddings on queries.
> > > >
> > > > We thank the reviewer for their clarifications. We believe there are several practical cases where our text conditioning method could be employed where multilabel classification would be insufficient.
> > > >
> > > > 1. **For use in retrieval augmented generation (RAG)** [1], particularly over a large knowledge pool. Candidate pools for RAG can span millions of facts, captions and labels. Multilabel image retrieval (without text conditioning) may return hundreds of valid candidates corresponding to dozens of aspects of the scene. This is too many to use in generation, are mostly unrelated to the user's goal, and potentially distracting to the LLM. Furthermore, in this case, the *user is already required to specify their information retrieval goals as query text (for use in the generation step)*.
> > > >
> > > > 2. **Scene level and conceptual queries.** Text queries can be used to create embeddings representing multiple objects and concepts taken together, which an object-level multilabel approach can't. An example from `CtrlBench`:
> > > >
> > > > **Q:** "What natural elements are most prominent in the foreground of this picture?"
> > > > **A:** "Silhouetted bare branches with some attached leaves crisscross the foreground, framing the view of the buildings"
> > > >
> > > > The user can control the embedding to represent **multiple** elements that are **natural** and positioned in the **image foreground**--in a single embedding. We believe that 1 & 2 would be difficult to accomplish with multilabel classification.
> > > >
> > > > Another important motivation is **accessibility**. Asking a question about an image and receiving a specific answer is intuitive, even for non-technical users. Whereas such users may find using a returned list of many labels more difficult.
> > > >
> > > > We will update the introduction to specifically reference our motivations and these use cases.
> > > >
> > > > >Even if CtrlBench truly contains image–text pairs with several valid answers, the ability to steer an embedding with a textual adapter does not guarantee that all valid facets are preserved; it only guarantees that one facet can be emphasised.
> > > >
> > > > As mentioned above, `CtrlBench` does contain problems that require using multiple conditions to emphasize multiple facets together. Furthermore, these facets can be combined arbitrarily from flexible text conditioning. Due to space constraints in figures, it is difficult to represent the more complex queries and display all the candidates. We agree that `CtrlBench` does not guarantee that all facets are preserved. However, `CtrlBench` does measure a model's ability to represent arbitrary combinations of facets. If the reviewer has time, clarification of the definition of "all valid facets" in this context would be useful.
> > > >
> > > > >Some minor details are still somewhat unclear; e.g. size of in-batch candidate pools for negative mining. One potential inconsistency that the author could perhaps resolve; the adapter is described as being added “after the frozen dual encoder”, yet the hooks are placed inside every decoder block of the LLM backbone?
> > > >
> > > >
> > > > For each image in the mini-batch we use 7 mined negative candidates, we mine negatives across the entire dataset. The vision encoder component of the VLLM is frozen. The LoRA hooks are in LLM component of the VLLM. We will revise the manuscript to clarify.
> > > >
> > > > >The new CLIP degradation table suggests 224-token CLIP is optimal, but it is still a little unclear how this was performed. Does that experiment rescale the entire image, rather than patch size vs. patch count trade-off that ABC exploits? If so, there is a worry that there may still be a discrepancy between baselines and the proposed model.
> > > >
> > > > **All methods scale visual tokens the same way.** The image is rescaled in all methods and the patch size (for example, 16x16 pixels) is always fixed. This is consistent across our method and the considered CLIP approaches. We will revise section 4.2 to clarify.
> > > >
> > > > -------------
> > > > [1] P. Lewis et al., “Retrieval-Augmented Generation for Knowledge-Intensive NLP Tasks,” 2020, doi: 10.48550/arxiv.2005.11401.

---

> > > > > ### Author Response · Authors · 2025-07-03
> > > > > **Thank you for your response! (2/2)**
> > > > >
> > > > > > The authors grant that leakage is a concern, and propose switching to OpenImages to source the dataset. However, it is not clear on what the authors mean when they say they will update reviewers on this task upon completion? Does this involve a complete restructuring on CtrlBench?
> > > > >
> > > > > Apologies, we should have been more clear. **Yes**, we have reconstructed CtrlBench. We settled on using the variant constructed using ADE20K [2] as we found the OpenImages variant had a lot of poor quality images (watermarked, etc.). **To the author's knowledge, no models evaluated below were trained on ADE20K.**
> > > > >
> > > > > | Model                | R@1   | R@5   | R@10  |
> > > > > |---------------------|-------|-------|-------|
> > > > > | **UniIR**           | 0.9 |  2.59  | 4.30 |
> > > > > | **MagicLens**       | 17.0     | 37.33     | 48.47     |
> > > > > | **VLM2Vec**         | 2.95 | 8.49 | 12.36 |
> > > > > | **GME-Qwen2VL-2B**  | 20.92 | 46.33 | 57.27 |
> > > > > | **GME-Qwen2VL-7B**  | 24.75 | 54.48 | 65.27 |
> > > > > | **ABC (pretrained)**| 11.71 | 32.48 | 42.69 |
> > > > > | **ABC (finetuned)** | **50.94** | **78.43** | **87.47** |
> > > > >
> > > > > We find that ABC still outperforms other approaches on an isolated visual distribution. The one difference in this variant is that we used an MLLM to create bounding boxes. We will add this result and a description of our construction pipeline to the manuscript.
> > > > >
> > > > > >If this is to be taken as a central contribution, re-sourcing the dataset from scratch during rebuttal seems unlikely to be achievable.
> > > > >
> > > > > We have been exploring options and developing tooling for recreating `CtrlBench` on different data distributions in advance of the rebuttal period. Mostly stemming from a frustration that a significant number of evaluation tasks are downstream from MSCOCO or ImageNet.
> > > > >
> > > > > >The authors provide an ablation table for a 500 step run, however the real model is quite a bit larger. More importantly, even if the property that "bidirectional loss is worse" even under scale holds, it seems that the authors do not demonstrate text-to-image results when both directions receive mined negatives. This is what we meant by asymmetry in our original comment. There is still some doubt whether the performance boost is an artefact of the experimental design.
> > > > >
> > > > > Thank you for clarifying the definition of asymmetry. We agree that exploring image negatives is an interesting direction. However, as images use more tokens than text, we found that training using image negatives was infeasible due to resource (GPU memory) constraints. We are happy to address this as a limitation.
> > > > >
> > > > > -------------
> > > > >
> > > > > [2] Scene Parsing through ADE20K Dataset. Bolei Zhou, Hang Zhao, Xavier Puig, Sanja Fidler, Adela Barriuso and Antonio Torralba. Computer Vision and Pattern Recognition (CVPR), 2017.

---

> > > > > > ### Author Response · Authors · 2025-07-04
> > > > > > **Updated Manuscript**
> > > > > >
> > > > > > Hello! We have updated the manuscript with the following changes:
> > > > > >
> > > > > > - Revised the introduction to include a use case for text conditioned visual retrieval (I.e. retrieval augmented generation).
> > > > > > - Amended to include requested details on mined negatives. Respectfully, we could not find the phrase "frozen dual encoder" in the manuscript.
> > > > > > - Updated section 4.2 to describe the exact method for scaling token budgets across baselines.
> > > > > > - Changed CtrlBench result to the version constructed using ADE20K to control for visual distribution.

---

### Review · Reviewer_xc7U · 2025-06-19

**Summary Of Contributions:**

The paper presents ABC model that uses natural language instructions to control visual embeddings by adapting
VLLMs into dense embedding models in a lightweight fashion with synthetic instructions. They also introduce
CtrlBench, a novel benchmark for measuring instruction-controlled retrieval on VQA tasks to highlight the interleave modalities for retrieving the correct response.

**Audience:**

Yes

**Broader Impact Concerns:**

N.A.

**Claims And Evidence:**

No

**Requested Changes:**

1. Clarification of the proposed method ABC against existing generative and non-generative LMMs, and how ABC functions on different downstream tasks in Introduction and Experimental section.
2. Include or revise the experimental design on the zero-shot retrieval and classification tasks to better reflect the claimed property on inverleaved multimodal embedding. For instance, classification/retrieval conditioned on text?
3. Comparison with sota generative LMMs on VQA task.

**Strengths And Weaknesses:**

A few comments below that might require revisions or clarifications:

1.	The proposed ABC as shown in Fig 2 involves a LLM in the training, but is it not operating as a generative large multimodal model (LMM) at testing? It is confusing how the model functions at inference time on different downstream tasks (classification, retrieval and VQA).
2.	I am not very convinced by the experimental design on the zero-shot retrieval and classification tasks in sec 4.2. The proposed method focuses on the inverleaved multimodal embedding to enable textual controllability on the visual embedding, but the experiments seem not reflecting this property.
3.	On the evaluation of VQA task in Sec 4.6, authors mention about retrieval. However, generative LMMs when evaluating on VQA task often do not involve retrieval but simply respond in an open-ended fashion or as multi-choice QA. Some clarification on the differences should be discussed and comparison against recent generative LMMs should be included, perhaps using tool suites, e.g.  https://github.com/EvolvingLMMs-Lab/lmms-eval

---

> ### Author Response · Authors · 2025-06-24
> **Thank you for your comments!**
>
> We thank the reviewer for their suggestions to further improve the paper. Please find our clarifications and proposed manuscript changes below.
>
> >A) Clarification of the proposed method ABC against existing generative and non-generative LMMs, and how ABC functions on different downstream tasks in Introduction and Experimental section.
>
> >B) The proposed ABC as shown in Fig 2 involves a LLM in the training, but is it not operating as a generative large multimodal model (LMM) at testing? It is confusing how the model functions at inference time on different downstream tasks (classification, retrieval and VQA).
>
> Our proposed method operates exclusively as a dense embedding model at inference time. *However,* we initialize the backbone using an MLLM before training. Our training process converts the features used by the MLLM for generative language modeling into features for crafting dense embeddings (for text-conditioned visual retrieval/classification). We will **add a background section** (Section 2) to the manuscript to clarify which tasks we consider and related work on converting generative models into embedding models.
>
> >C) Include or revise the experimental design on the zero-shot retrieval and classification tasks to better reflect the claimed property on inverleaved multimodal embedding. For instance, classification/retrieval conditioned on text?
>
> >D) I am not very convinced by the experimental design on the zero-shot retrieval and classification tasks in sec 4.2. The proposed method focuses on the inverleaved multimodal embedding to enable textual controllability on the visual embedding, but the experiments seem not reflecting this property.
>
> Confusingly, multimodal embedding literature has adopted the term "Visual Question Answering" (VQA) to refer to text-conditioned visual classification/retrieval tasks [1,2,3]. These tasks use dense embeddings of queries (images and text instructions) to retrieve from a fixed pool of candidates, such as class labels or captions. **Table 4 and our own benchmark (table 5) evaluate a variety of text-conditioned visual retrieval/classification tasks, NOT open-ended VQA like [4].** We do not evaluate any open-ended generation tasks. The objective of tables 1,2,3 is to validate the performance of our pretrained model, whereas tables 4,5 evaluate the text-conditioned adapter. **We will clarify this terminology in the added background section.**
>
> [1] Jiang et al., "VLM2Vec: Training Vision-Language Models for Massive Multimodal Embedding Tasks." The Thirteenth International Conference on Learning Representations, 2025.
> [2] Zhang et al., "GME: Improving Universal Multimodal Retrieval by Multimodal LLMs." arXiv preprint, 2024.
> [3] Wei et al., “UniIR: Training and Benchmarking Universal Multimodal Information Retrievers.” in Computer Vision - ECCV 2024, vol. 15145, 2024.
> [4] Antol et al., “VQA: Visual Question Answering.” in 2015 IEEE International Conference on Computer Vision (ICCV), IEEE, 2015.

---

> > ### Author Response · Authors · 2025-06-29
> > **A couple additional clarifications**
> >
> > > Table 4 and our own benchmark (table 5) evaluate a variety of text-conditioned visual retrieval/classification tasks
> >
> > We provide an example from a "VQA" task from the Visual7W split in figure 6. The VQA task uses an image conditioned on text (phrased as a question) to classify the location of the image. It determines the answer "a beach" by comparing the similarity of the joint image-question embedding with embeddings of text candidates. We provide examples from CtrlBench in Appendix F. We note that although we only provide examples of several candidates in these figures, the actual task retrieves from thousands of options.
> >
> > > Comparison with sota generative LMMs on VQA task.
> >
> > Due to their open-ended answering, LMMs cannot directly be applied to the tasks we evaluate. Our evaluations require retrieving the best option from a fixed pool of candidates. Some additional algorithm or model would be required applied to map the open-ended LMM output to our candidate pool.
> >
> > We have clarified the manuscript to provide background on our setting and are happy to answer additional questions.

---

> > > ### Author Response · Authors · 2025-07-02
> > > **Follow-up on revisions**
> > >
> > > Hello! Have our clarifications and manuscript revisions (especially the addition of the background section) addressed your concerns? We are happy to answer any additional questions.

---

### Review · Reviewer_sjVm · 2025-06-20

**Summary Of Contributions:**

This paper proposes a method to achieve multimodal embeddings (visual embeddings made of image + text instructions, where visual embeddings can be guided and controlled by natural language instructions). Unlike CLIP which cannot handle tasks that require user instruction or disambiguation (e.g. distinguishing between two plausible image captions), this paper proposes to solve this issue by plugging in a Large VLM (Qwen2-VL-7B) into the embedding process. Unlike CLIP, VLMs are unified image-text model with early fusion that integrate image and text instructions to build multimodal embeddings. The authors modify the attention to bi-directional attention and remove the generative part of the VLM and finetune it. Specifically, there are two stages: 1) pretraining with constative learning on image-text pairs with hard negative mining of the text pairs (captions that achieve score just lower than the threshold in a preminlaty stage of training), 2) fine-tuning on instruction data gathered from Visual Genome and using GPT-4 to reformulate these to instruction-retrieval format, using also contrastive training with in-batch negatives (negatives of the same image but of different regions, as well as from other samples in the batch). Both pretraining and fine-tuning use separate LoRA adaptors.  Experiments are conducted to demonstrate the method’s raw embedding quality and its ability to handle instruction-guided queries (zero-shot image/text retrieval, zero-shot image classification, visual question answering (VQA), and a new proposed instruction-controlled retrieval benchmark).

**Audience:**

Yes

**Broader Impact Concerns:**

No issues

**Claims And Evidence:**

Yes

**Requested Changes:**

I am worried mostly about the practicability of the method, as 1) i do not see its advantage over autoregressive VLMs (which the authors use as a backbone and finetune) and using its embeddings directly, and 2) its increased computation time and memory. There are other weaknesses as mentioned.

**Strengths And Weaknesses:**

Strengths
- Clever engineering in the method
- Impressive improvements over baselines

Weaknesses:
- My biggest problem is why do we need specialized multimodal embedding models, if a large autoregressive VLM (trained to generate text, which the authors use as a backbone) can achieve this purpose? Why not use the embeddings from this model itself, without any further training/tuning? What is wrong with those standalone embeddings? These models already already change and control the image embeddings according to the instruction (due to prefix-LM where the image tokens and instructions tokens attend to each other). So natural language guided visual embeddings are already achieved by VLMs. This is the main question which is not addressed. There is also no comparison in the tables, with using those standalone embeddings from those VLMs directly. Why do we even need a multimodal embedding model to solve tasks such as VQA for example, if a text generative VLM can do this?
- The improvements are not surprising, because of the use of a powerful new 7B VLM as an additional embedding model. This increases the memory requirements largely. Additionally, there is a huge amount of computation involved for this improvement in performance. I do not find that as a clever way of achieving a goal, when a huge new model is plugged in for that purpose. Moreover, unlike CLIP, there should be two runs on the same model for every sample. One with the pretreated model from stage 1 to get the embeddings for the answer candidates, and the other with the fine-tuned model to get the image+instruction embedding.
- The approach in spirit is similar to VLM2Vec. Therefore, the novelty is limited. No comparisons with VLM2Vec in Tables 1,2,4.
- No ablation studies on the hard negative mining in the pretreating stage. The improvements could be largely coming from the strong hard negative mining.
- Are the results in the Tables using an image size of 224x224? Otherwise, they are not a fair comparison since the authors use Qwen-VL which can accept different image size. Most baselines use an image size of 224x224, while the authors use a larger image size?
- The related work section does not mention how the approach is advantageous over current methods, which is the most important part of the related work. It simply states other works without describing how the proposed approach differs.

Minor
- In section 3, “uses self-supervised training on image-caption pairs” - image captioning is not a supervised task

---

> ### Author Response · Authors · 2025-06-24
> **Thank you for your comments! (part 1/2)**
>
> Thank you for your valuable comments and feedback.
>
> >My biggest problem is why do we need specialized multimodal embedding models, if a large autoregressive VLM (trained to generate text, which the authors use as a backbone) can achieve this purpose? Why not use the embeddings from this model itself, without any further training/tuning? What is wrong with those standalone embeddings? These models already already change and control the image embeddings according to the instruction (due to prefix-LM where the image tokens and instructions tokens attend to each other). So natural language guided visual embeddings are already achieved by VLMs. This is the main question which is not addressed. There is also no comparison in the tables, with using those standalone embeddings from those VLMs directly. Why do we even need a multimodal embedding model to solve tasks such as VQA for example, if a text generative VLM can do this?
>
> We agree that this is an important baseline to address. Throughout our experiments we have found embeddings extracted directly from a large autoregressive VLM perform poorly, even on relatively simple tasks like matching images to captions (MSCOCO):
>
> | Pooling Method | Direction | Top-1  | Top-5  | Top-10 |
> |----------------|-----------|--------|--------|--------|
> | Mean Pooling   | i2t       | 6.30%  | 16.18% | 23.70% |
> |                | t2i       | 3.41%  | 9.25%  | 13.39% |
> | Last Token     | i2t       | 0.06%  | 0.26%  | 0.36%  |
> |                | t2i       | 0.08%  | 0.30%  | 0.73%  |
>
> We experimented with extracting a dense embedding using both the last token and mean pooling and found that neither has satisfactory performance (although mean pooling performs significantly better than last token). We will update the manuscript to include these results and motivate the need for additional training.
>
> >The approach in spirit is similar to VLM2Vec. Therefore, the novelty is limited. No comparisons with VLM2Vec in Tables 1,2,4.
> No ablation studies on the hard negative mining in the pretreating stage. The improvements could be largely coming from the strong hard negative mining.
>
> Comparing with VLM2Vec directly is difficult as it is trained directly on the majority of tasks in Tables 1,2,4. However, we do provide this comparison in Appendix D. We find that VLM2Vec tends to outperform on tasks used in its training data and underperform ABC on tasks where they are both out-of-distribution. Ablating over the use of our mined negatives during pretraining, we find that our negative mining regime increases performance in both i2t and t2i retrieval:
>
> | # Negatives | MSCOCO i2t | MSCOCO t2i |
> |-------------|-----------|-----------|
> | 0           | 50.42    | 35.71    |
> | 3           | 58.42    | 43.97    |
> | 6           | 59.94    | 44.84    |
>
> pretraining 500 steps with a batch size of 256.
>
> However, additional adaptation to use text-instructions is required, as the pretrained model has very poor performance on CtrlBench:
>
> | Model           | R@1   | R@5   | R@10  |
> |-----------------|-------|-------|-------|
> | UniIR           | 0.0   | 0.0   | 0.1   |
> | MagicLens       | 11.7  | 28.4  | 38.5  |
> | VLM2Vec         | 26.44 | 56.5  | 68.9  |
> | GME-Qwen2VL-2B  | 9.4   | 24.6  | 33.9  |
> | GME-Qwen2VL-7B  | 10.7  | 27.6  | 36.9  |
> | **ABC (pretrained)**  | **5.73** | **15.6** | **22.53** |
> | **ABC (finetuned)**  | **41.0** | **56.5** | **80.0** |
>
> We will update the manuscript to include the ablation over our negative mining regime and pretrained baseline on CtrlBench.
>
> > The improvements are not surprising, because of the use of a powerful new 7B VLM as an additional embedding model. This increases the memory requirements largely. Additionally, there is a huge amount of computation involved for this improvement in performance. I do not find that as a clever way of achieving a goal, when a huge new model is plugged in for that purpose. Moreover, unlike CLIP, there should be two runs on the same model for every sample. One with the pretreated model from stage 1 to get the embeddings for the answer candidates, and the other with the fine-tuned model to get the image+instruction embedding.
>
> We agree that increased computational/resource demand requirements is a limitation of our method. However, we consistently found that similar approaches that utilized only a pretrained CLIP backbone (Uniir, MagicLens) had very poor performance on any tasks that required utilizing instructions (tables 4 and 5). Where possible we try to compare to similar-sized baselines such as VLM2Vec and GME. However, this is difficult due to their training distributions being overlapped with testing distributions (as discussed above). In a practical setting, the efficiency of embedding models greatly benefits from caching. I.e. a distinct query or candidate can be inferred once and cached indefinitely in a vector database.

---

> > ### Author Response · Authors · 2025-06-24
> > **Thank you for your comments! (part 2/2)**
> >
> > >Are the results in the Tables using an image size of 224x224? Otherwise, they are not a fair comparison since the authors use Qwen-VL which can accept different image size. Most baselines use an image size of 224x224, while the authors use a larger image size?
> >
> > We initially attempted to scale up all related work to have comparable token budgets. However, we found that certain baselines degraded instead of improving when scaling token budgets. In particular, CLIP models consistently degrade when scaling tokens budgets beyond their pretraining resolution. See MSCOCO t2i and i2t results for OpenClip-ViT-g-14 below:
> >
> > | Resolution | i2t Top-1 | t2i Top-1 |
> > |------------|-----------|-----------|
> > | 224x224    | 64.94    | 48.02    |
> > | 336x336    | 63.28    | 47.42    |
> > | 448x448    | 61.98    | 45.97    |
> >
> > Therefore, we constrain the token budgets of CLIP models to performance optimal settings during evaluations. When evaluating baselines designed to have adaptive token budgets (such as VLM2Vec [1] and GME [2]), we scale token budgets to match our method, accordingly. We will revise Section 4 to discuss our choice of inference settings for baselines and include the above result.
> >
> > > The related work section does not mention how the approach is advantageous over current methods, which is the most important part of the related work. It simply states other works without describing how the proposed approach differs.
> >
> > We will update the related work to describe our advantages over previous work, such as **1)** our method requires much less instruction finetuning data and has a flexible adapter design. **2)** No pretraining on large-scale human annotated data sources, such as MSCOCO.
> >
> > > In section 3, “uses self-supervised training on image-caption pairs” - image captioning is not a supervised task
> >
> > Thank you for catching this error, we will fix it in the updated manuscript.
> >
> > [1] Jiang et al., "VLM2Vec: Training Vision-Language Models for Massive Multimodal Embedding Tasks." The Thirteenth International Conference on Learning Representations, 2025.
> > [2] Zhang et al., "GME: Improving Universal Multimodal Retrieval by Multimodal LLMs." arXiv preprint, 2024.

---

> > > ### Comment · Reviewer_sjVm · 2025-06-29
> > > **Response to authors**
> > >
> > > I thank the authors for their response and for the extra experiments. My concerns are now cleared and it is great to see the revisions in the manuscript. I have no more further remarks and i am now positive about the work.

---

### Author Response · Authors · 2025-06-29
**Summary of Revisions**

We thank all the reviewers for their time and valuable feedback, which we believe strengthened our paper. We have revised our paper to address  suggestions, as promised in our rebuttal.

We highlight all changes in blue and are happy to address any further feedback or revisions.
Below is a brief summary of the revisions we have made.

-------------

# Main Paper

We reorganized the paper to include a background section (section 2) and include related work with the discussion (section 5).

**Introduction**
- (R3) Updated the introduction and Figure 1 with an example of ambiguous visual retrieval from CtrlBench.

**Background**
- (R2) Added background on converting generative language models into dense embedding models.
- (R2) Clarified the definition of VQA for embedding models.
- (R3) Defined and discussed ambiguity.

**Method**
- (R3) Clarified the method to include the requested details from W4.
- (R1) Updated to remove misuse of term "self-supervised".

**Experiments**
- (R1, R3) Added a section on inference and resolution settings (section 4.2) and results on scaling token budgets for CLIP models.
- (R1, R3) Added negative mining i2t and t2i ablation, which demonstrates that our negative mining method strengthens the pretrained model.
- (R1, R3) Added a baseline of ABC before finetuning on CtrlBench.
- (R3) Clarified definition of mined negatives vs. randomized negatives and updated section 4.3 to remove the claim that our weaker t2i performance results from our negative mining regime.

**Discussion and Related Work**
- (R1) Updated related work to discuss the advantages of our method compared to prior work. (section 5.3)

**Broader Impact Statement**
- (R3) Added a discussion of the potential positive and negative impacts of our work.

# Appendix
- (R1) Added results on extracting dense embeddings from VLLMs without contrastive training.

---

### Decision · Action_Editor_YjHn · 2025-07-17

**Recommendation:** Accept as is

**Audience:**

Yes

**Audience Explanation:**

Yes, researchers working on the multimodal field can be interested in the paper.

**Claims And Evidence:**

Yes

**Claims Explanation:**

This paper proposes a method to achieve multimodal embeddings, that is, visual embeddings made of image + text instructions, where visual embeddings can be guided and controlled by natural language instructions. This is achieved via plugging in a Large VLM (Qwen2-VL-7B) into the embedding process. The paper received very detailed review feedback. After rebuttal, the recommendations were Accept, Leaning Accept, and Leaning Reject.

On one hand, reviewers mentioned that this paper has clever engineering in the method and impressive improvements over baselines. The proposed CtrlBench is also interesting.

On the other hand, reviewers also had several shared concerns. For example, the motivation of the paper is unclear, and the initial reading of the paper has led to some confusions. This is mostly due to terminologies that have been used in different context carrying different meaning and are associated with different experimental protocols. Furthermore, the comparison with CLIP regarding the use of image resolution is unfair. The design of the original CtrlBench may have data leakage issue.

The authors have made a genuine effort during rebuttal, adding a background section, clarifying many hyper-parameters, rebuilding CtrlBench on ADE20K, and inserting a brief ethics note. Two reviewers mentioned that their initial concerns are cleared in the rebuttal and the paper's revision look a lot better. However, one reviewer still showed concerns regarding the motivation of the paper, resolution confounder, and the new CtrlBench v2 benchmark introduced during rebuttal.

The AC acknowledges the significant effort invested in the rebuttal and the resulting improvements, and thinks that the claims are generally supported by clear and convincing evidence, and that the revisions made during the rebuttal have effectively addressed the reviewers' points of confusion. Therefore, the AC would like to recommend acceptance by the end.